# Deciphering the heterogeneity of the Lyve1⁺ perivascular macrophages in the mouse brain

C. Siret[1], M. van Lessen[2], J. Bavais[1,3], H. W. Jeong [4], S. K. Reddy Samawar[5], K. Kapupara [5], S. Wang[1], M. Simic[1], L. de Fabritus[1], A. Tchoghandjian[6], M. Fallet[1], H. Huang[1,7], S. Sarrazin[1], M. H. Sieweke [1,7], R. Stumm [8], L. Sorokin [5], R. H. Adams [4], S. Schulte-Merker[2], F. Kiefer[4,9] & S. A. van de Pavert [1] ✉

Perivascular macrophages (pvMs) are associated with cerebral vasculature and mediate brain drainage and immune regulation. Here, using reporter mouse models, whole brain and section immunofluorescence, flow cytometry, and single cell RNA sequencing, besides the Lyve1⁺F4/80⁺CD206⁺CX3CR1⁺ pvMs, we identify a CX3CR1⁻ pvM population that shares phagocytic functions and location. Furthermore, the brain parenchyma vasculature mostly hosts Lyve1⁺MHCII⁻ pvMs with low to intermediate CD45 expression. Using the double *Cx3cr1^GFP* x *Cx3cr1-Cre;Rosa^{tdT}* reporter mice for finer mapping of the lineages, we establish that CD45^low CX3CR1⁻ pvMs are derived from CX3CR1⁺ precursors and require PU.1 during their ontogeny. In parallel, results from the *Cxcr4-CreErt2;Rosa26^{tdT}* lineage tracing model support a bone marrow-independent replenishment of all Lyve1⁺ pvMs in the adult mouse brain. Lastly, flow cytometry and 3D immunofluorescence analysis uncover increased percentage of pvMs following photothrombotic induced stroke. Our results thus show that the parenchymal pvM population is more heterogenous than previously described, and includes a CD45^low and CX3CR1⁻ pvM population.

The central nervous system (CNS) has long been considered immune privileged, devoid of immune cells other than microglia and lacking classical lymphatic vessels. The re-discovery of the dural lymphatic network highlights a route for the drainage of the brain to the periphery[1–7]. Recent MRI data indicate that drainage from the CNS parenchyma occurs towards lymphatics and draining lymph nodes via the foramen[7].

Besides microglia, several myeloid populations have now been characterized, shown to be essential for brain homeostasis and to contribute to brain diseases[8]. These cells include non-parenchymal or border-associated macrophages (BAMs), which can be subclassified into perivascular, subdural meningeal, and choroid plexus macrophages. These macrophages are established during development by embryonic precursors derived from the yolk sac and with the exception of choroid plexus macrophages, are not replaced by blood monocytes during adulthood[9,10]. Specifically, perivascular macrophages (pvMs) are located in the perivascular space of the blood vessel, delimited by the vascular basement membrane of blood vessels

[1]Aix-Marseille Univ, CNRS, INSERM, Centre d'Immunologie de Marseille-Luminy (CIML), Marseille, France. [2]Institute for Cardiovascular Organogenesis and Regeneration, Faculty of Medicine, University of Münster, Münster, Germany. [3]Turing Centre for Living systems, Marseille, France. [4]Max Planck Institute for Molecular Biomedicine, Münster, Germany. [5]Institute of Physiological Chemistry and Pathobiochemistry and Cells in Motion Interfaculty Centre (CIMIC), University of Münster, Münster, Germany. [6]Aix-Marseille Univ, CNRS, INP, Inst Neurophysiopathol, Marseille, France. [7]Center for Regenerative Therapies Dresden (CRTD), Technische Universität Dresden, Dresden, Germany. [8]Institute of Pharmacology and Toxicology, Jena University Hospital, Jena, Germany. [9]University of Münster, European Institute for Molecular Imaging (EIMI) and Cells in Motion Interfaculty Centre (CIMIC), Münster, Germany. ✉e-mail: vandepavert@ciml.univ-mrs.fr

and the glial basement membranes[11]. The pvMs have been shown to be involved in many processes in the CNS. Early studies demonstrated that pvMs are capable of scavenging molecules injected into the cerebral ventricles. They were proposed to be involved in blood–brain barrier function and mediate the uptake of macromolecules[12–14], and participate in immune regulation[15]. Data from animal models indicate that these cells are involved in a wide variety of brain related disorders such as cerebrovascular and neurocognitive functions and blood–brain barrier functioning in hypertension[16,17], brain infections, immune activation, Alzheimer's disease[18–20], and multiple sclerosis[21,22], suggesting that they are a key component of the brain-resident immune system and involved in clearing or draining of the CNS.

The Lymphatic Vessel Endothelial Hyaluronan Receptor (Lyve1) is notably expressed on lymphatic endothelial cells (LEC) and selected macrophages, but not on microglia. In the periphery, macrophage populations lining the blood vessels or nerve bundles have been characterized as Lyve1hi and Lyve1lo, respectively[23]. In the mouse brain, myeloid cells are a heterogeneous group of cells localized in specific niches and include parenchymal microglia and non-parenchymal pvMs[8,24]. Previously, brain pvMs were described to be Lyve1+CD45high and were shown to express canonical macrophage markers such as fractalkine receptor (CX3CR1), colony-stimulating factor1 receptor (Csf-1R), CD206, and Iba-1, but also the prototypic macrophage markers CD11b and F4/80[8,9,25,26].

Recent studies have characterized pvMs and BAM within and outside the CNS, including the peripheral nervous system[23,27,28]. Particularly, parenchymal Lyve1+ pvMs were shown to be heterogenous by immunofluorescence on sections[25].

Here we show, using different reporter and fate-mapping mouse models and employing (whole-mount) immunofluorescence as well as flow cytometry and single-cell RNA sequencing, that the Lyve1+ pvM population in the mouse brain parenchyma is more heterogenous than previously described, and that Lyve1+F4/80+CD206+ pvMs include a previously missed CX3CR1−CD45low population within the parenchyma. Moreover, most parenchymal pvMs lacked MHC-II expression. Using multiple lymphatic reporter models, we exclude the presence of "bona fide" lymphatic endothelial cells within the brain parenchyma, thereby demonstrating that these parenchymal CD45lowLyve1+ cells are not lymphatic endothelial cells. Functionally, these unconventional pvMs are able to phagocytose macromolecules injected into the ventricles. During photothrombotic-induced stroke, pvM populations increase in numbers, without recruitment of cells from bone marrow. Hence, our data describe the heterogeneity of parenchymal pvMs and identify a previously overlooked non-conventional CX3CR1− pvM population.

## Results

### Identification of a Lyve1+CX3CR1− cell population in the brain parenchyma

Using whole-mount immunofluorescence of the cortex at the dorsal and ventral sides and the cerebellum, we identified Lyve1+ cells adjacent to blood vessels that did not form lumenized vessels (Figs. 1a–d, 3D whole brain in Supplementary Movie 1 and maximum intensity projection in Supplementary Fig. 1a). Morphology of these cells depended on their location. Within the dorsal cortex (Fig. 1a), Lyve1+ cells were small and spread, similar to macrophages in the pia mater. Within the ventral cortex, these cells were elongated and more stretched, as Lyve1+ cells in the hippocampus (Fig. 1b, c). In the olfactory bulb, near the cribriform plate, where cerebrospinal fluid (CSF) lymphatic drainage occurs from the sub-arachnoid space into nasal lymphatics[7], numerous parenchymal Lyve1+ cells were also found to be long and stretched (Fig. 1d). Analysis of brain sections of adult *Cx3cr1GFP/+* mice revealed that most Lyve1+F4/80+Iba1+CD206+ pvMs expressed as expected the CX3CR1 chemokine receptor[9] (white arrows in Fig. 1e–g). However, in addition, we observed a rare population of CX3CR1−Lyve1+F4/80+Iba1+CD206+ pvMs that also display low expression of CD45 (Fig. 1h).

Notably, almost all Lyve1+ cells lacked MHC-II expression (red arrow in Fig. 1i), and only few Lyve1−MHC-II+ cells were observed (pink arrows in Fig. 1i). We confirmed lack of CX3CR1GFP expression in this Lyve1+ pvM population in the *Cx3cr1GFP/+* adult mouse brain by staining sections with anti-GFP (Supplementary Fig. 1b).

### Lyve1+CX3CR1− cells are non-conventional parenchymal perivascular macrophages

In the light of the recent re-discovery of lymphatic vessels in the meningeal compartment[1,6,29], we investigated whether the Lyve1+CX3CR1− population in the parenchyma of adult mice was of lymphatic origin. We observed parenchymal Lyve1+ cells in very close association with blood vessels (Fig. 2a–d, Supplementary Movie 2) outside of a structure called "pia mater cul de sac" (Fig. 2b–d). The pia mater co-migrates with the penetrating arteries into the brain parenchyma during development[30], visualized by podoplanin[11], surrounding arterioles within the cortex. We observed that the podoplanin+ pia was located between the vascular endothelium and the Lyve1+ cells (Fig. 2b–d, Supplementary Fig. 2a). We also observed that the Lyve1+ cells were positioned within a laminin γ1+ or laminin α1+ basement membrane within the perivascular space[31] (Figs. 2e–i, 3D reconstruction movie showing CX3CR1+ and CX3CR1− cells and their nuclei in Supplementary Movies 3–4). This compartment is defined by the vascular basement membrane on the abluminal side of the vessel wall and by the parenchymal basement membrane on the CNS side[32]. We observed that parenchymal Lyve1+ cells did not express CD31 (Fig. 2a–k), podoplanin (Fig. 2b–d), VEGFR3 (Fig. 2j, k) or the master regulator for lymphatic endothelial identity Prox1 (Fig. 2l–o). We verified the non-lymphatic phenotype using *Prox1mOrange2* and *Prox1-CreErt2;Rosa26tdT* reporter mice[33,34] (Supplementary Fig. 2b, c). We analyzed the *Prox1-CreErt2;Rosa26tdT* brain, 2 weeks after Tamoxifen injection, and did not observe *Prox1+* lymphatic endothelial cells, ruling out any LEC identity within the brain parenchyma. While the dura mater contains lymphatic vessels, no conventional lymphatic vessels have been described within the parenchyma or in the leptomeninges. Similarly, while we did not observe lumenized lymphatic vessels within the pia mater, we observed single Lyve1+Prox1+ lymphatic endothelial cells in the pia mater (Supplementary Fig. 2d) as was recently described for mammals[35].

To further rule out an astrocyte, glia, fibroblast, or pericyte identity of the Lyve1+CX3CR1− cells, we analyzed immunofluorescence staining for aquaporin-4 (AQP4) and GFAP (Supplementary Fig. 2e), ER-TR7 (Supplementary Fig. 2f), PDGFRβ (Supplementary Fig. 2g) and observed the parenchymal Lyve1+ cell population was negative for all these markers. Furthermore, we excluded a neural crest cell origin using the *Wnt1-Cre; Rosa26tdT* reporter mice (Supplementary Fig. 2h).

We confirmed the presence of the parenchymal CX3CR1− Lyve1+ populations in neonatal up to 1-year-old mice by flow cytometry. Using *Cx3cr1GFP* brain parenchyma cell suspension devoid of meninges, conventional pvMs were identified as CD64+CD206+F4/80+CX3CR1+ (Fig. 3a, gating strategy in Supplementary Fig. 3a). The CD64+CD206+F4/80+ population was both Ly6C− and Ly6G−, thus excluded monocytes, dendritic cells or neutrophils (Supplementary Fig. 3a). Microglia were Lyve1−CD11b+ and thus not part of the Lyve1+F4/80+CD11b+ populations (Supplementary Fig. 3b). At 2 days after birth (P2) we observed a small CD64+F4/80+CD206+CX3CR1− population which increased at P7, 14, and 21 (Fig. 3a, b) and peaked in adults. Subsequently, we observed a low percentage in 1-year old brain (Fig. 3a, relative CD64+F4/80+CD206+CX3CR1− cell numbers in Fig. 3b). In order to further characterize the parenchymal CD64+F4/80+CD206+CX3CR1− population, we analyzed their expression of macrophage markers compared to the CX3CR1+ population (Fig. 3a) and of microglia (Supplementary Fig. 3b). Notably, the CD64+F4/80+CD206+CX3CR1− population expressed CD45 at low levels. Other known macrophage markers, such as Lyve1 and CD163 were expressed

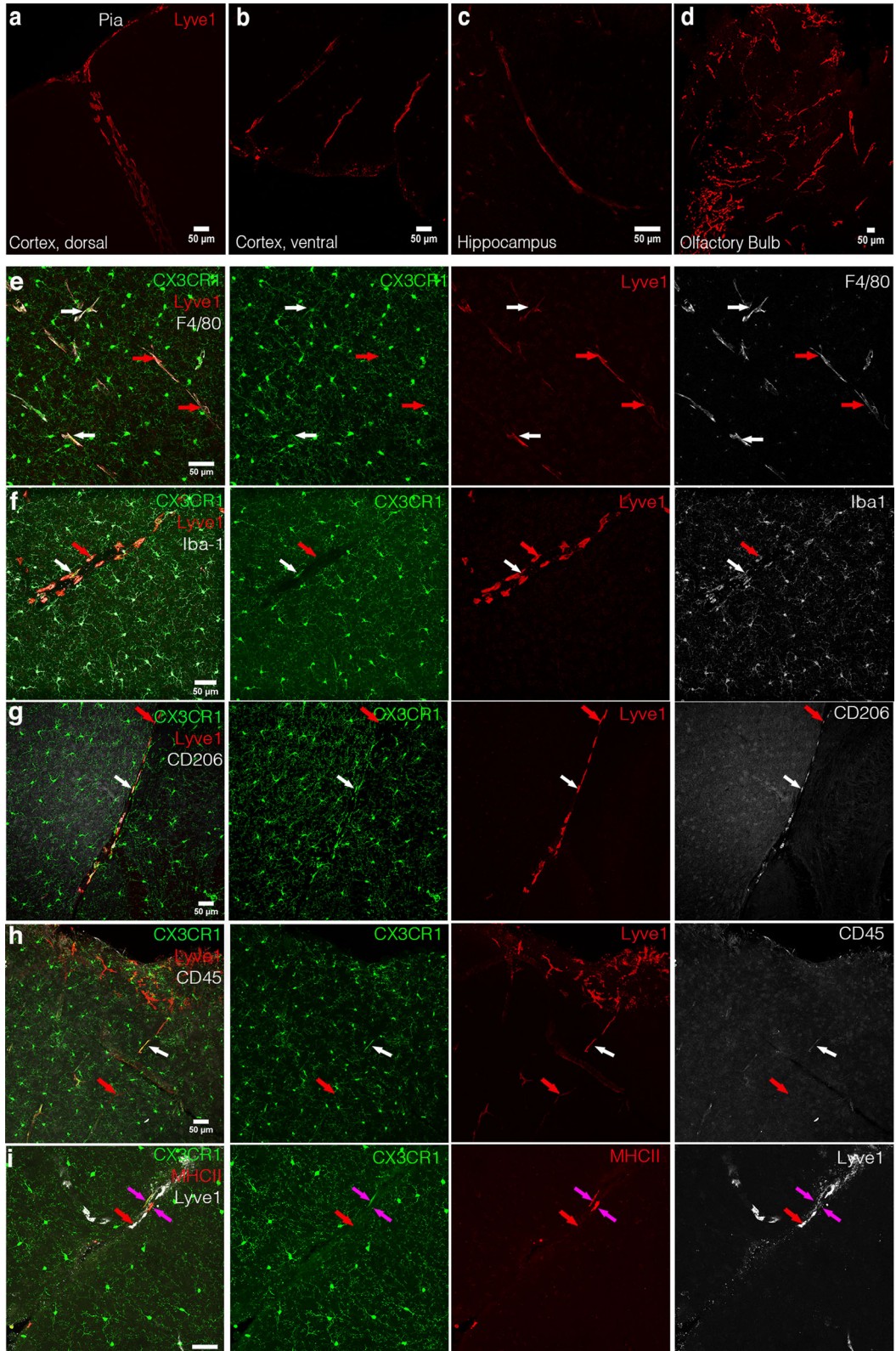

**Fig. 1 | Identification of a Lyve1⁺CX3CR1⁻ population in the mouse brain.** Confocal microscopy of brain sections stained with anti-Lyve1 showing the different Lyve1⁺ cell morphologies in the cortex at the dorsal side (17 μm maximum intensity projection) (**a**), in the cortex at the ventral side (27 μm maximum intensity projection) (**b**), the Hippocampal fissure within the hippocampus (15 μm maximum intensity projection) (**c**), and in the olfactory bulb (61 μm maximum intensity projection) (**d**) of an adult mouse brain. **e**–**i** Immunofluorescence microscopy on sections of *Cx3cr1^GFP* in the dorsal cortex of the mouse brain. **e** Staining of *Cx3cr1^GFP* sections for Lyve1 (red) and F4/80 (white) (50 μm maximum intensity projection), **f** Iba1 (29 μm maximum intensity projection), **g** CD206 (20 μm maximum intensity projection), **h** CD45 (46 μm maximum intensity projection), **i** MHC-II (18 μm maximum intensity projection). White arrows point to conventional Lyve1⁺CX3CR1⁺ pvMs. The red arrows indicate Lyve1⁺CX3CR1⁻ cells. The pink arrows indicate Lyve1⁻MHCII⁺ cells. (*n* = 4).

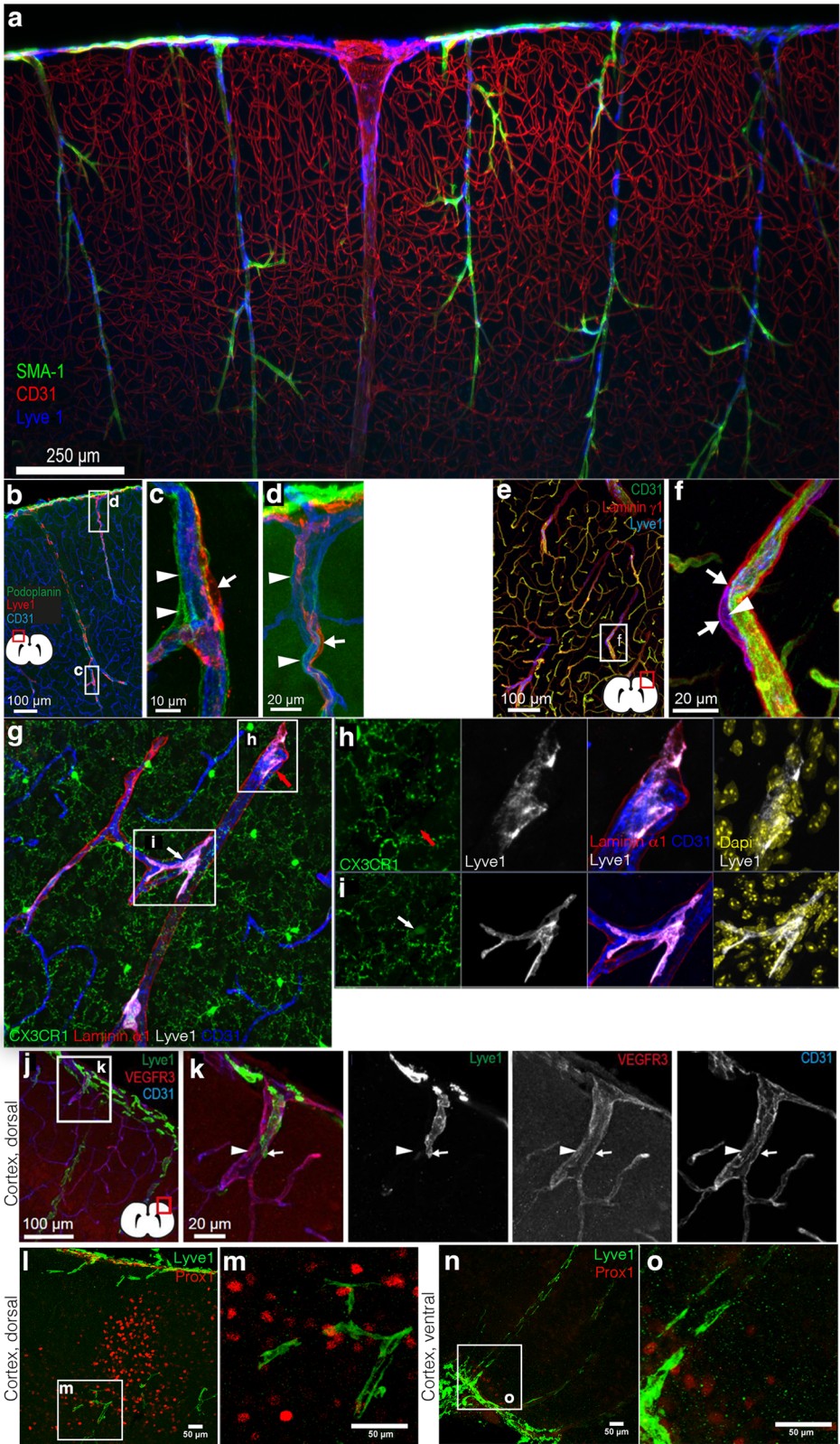

at similar levels as the CD64⁺CD206⁺F4/80⁺CX3CR1⁺ population (Fig. 3a), indicating a macrophage identity of the CD64⁺F4/80⁺CD206⁺CX3CR1⁻ population. To further confirm the macrophage phenotype, we sorted the parenchymal Lyve1⁺F4/80⁺CX3CR1⁺ and Lyve1⁺F4/80⁺CX3CR1⁻ populations and determined that these populations had a macrophage morphology[36] (Fig. 3c).

Collectively, the cytometry profile plus their morphology confirmed that the parenchymal perivascular macrophage is heterogenous

with respect to CX3CR1 expression, and that the CX3CR1⁻ population is a bona-fide macrophage population.

## Gene expression profiling of the pvM populations

To molecularly define the pvM subpopulations, we characterized their transcriptomes by single-cell RNA sequencing of the non-neuronal cell population in the cortex of juvenile (10 days), adult (7–11 weeks) and aged (1.5 years) mice (GSE133283) (Fig. 4a). The datasets of the three

**Fig. 2 | Perivascular Lyve1⁺ cell location. a** 200 μm Maximum intensity projection of a lightsheet microscope acquisition of the cortex at the dorsal side in a cleared mouse brain labeled for SMA (green), CD31 (red), and Lyve1 (blue) (*n* = 2).
**b** Confocal microscopy of brain sections stained for podoplanin (green), Lyve1 (red), and CD31 (blue) (*n* = 4), inserts (**c**) and **d** zoom in on Lyve1⁺ cells lining the blood vessels and placed outside of the « cul-de-sac of the pia mater » defined by the podoplanin labeling. **e** Confocal microscopy of a dorsal cortex section stained for CD31 (green), laminin-γ1 (red), and Lyve1 (blue) (*n* = 3) with insert (**f**) showing a higher magnification showing the Lyve1⁺ cells in the perivascular space (*n* = 2).
**g** 55 μm cryosection of the *Cx3cr1ᴳᶠᴾ* brain cortex, with the location of Lyve1⁺(white) pvMs lining the blood vessel (CD31, blue) but within the perivascular space delineated by the laminin α1 staining (red). **h** CX3XR1⁺ pvMs (red arrow), while **i** is a

CX3XR1⁺ pvMs (white arrow). Higher resolutions of the inserts are shown to the right, with DAPI (yellow). 3D reconstructions of these cells highlighting the nuclei present within the cells are presented in Supplementary Movies 3 (**h**) and 4 (**i**) (*n* = 2). **j** Lyve1⁺ cell characterization by immunofluorescence on adult brain sections for Lyve1 (green), VEGFR3 (red), and CD31 (blue) staining. **k** The arrowhead points at a blood vessel and the white arrow shows a Lyve1⁺ cell negative for VEGFR3 and CD31. **l** Anti-Prox1 staining of a dorsal cortical brain area (39 μm maximum intensity projection). **k** high magnification of the Lyve1⁺ and Prox1 cell in **l** (*n* = 1). **n** Also in the ventral cortex Lyve1⁺ cells (green) did not express Prox1 (red). **o** High magnification from area in **n** showing the individual Lyve1 cells without Prox1 expression (50 μm maximum intensity projection) (*n* = 2).

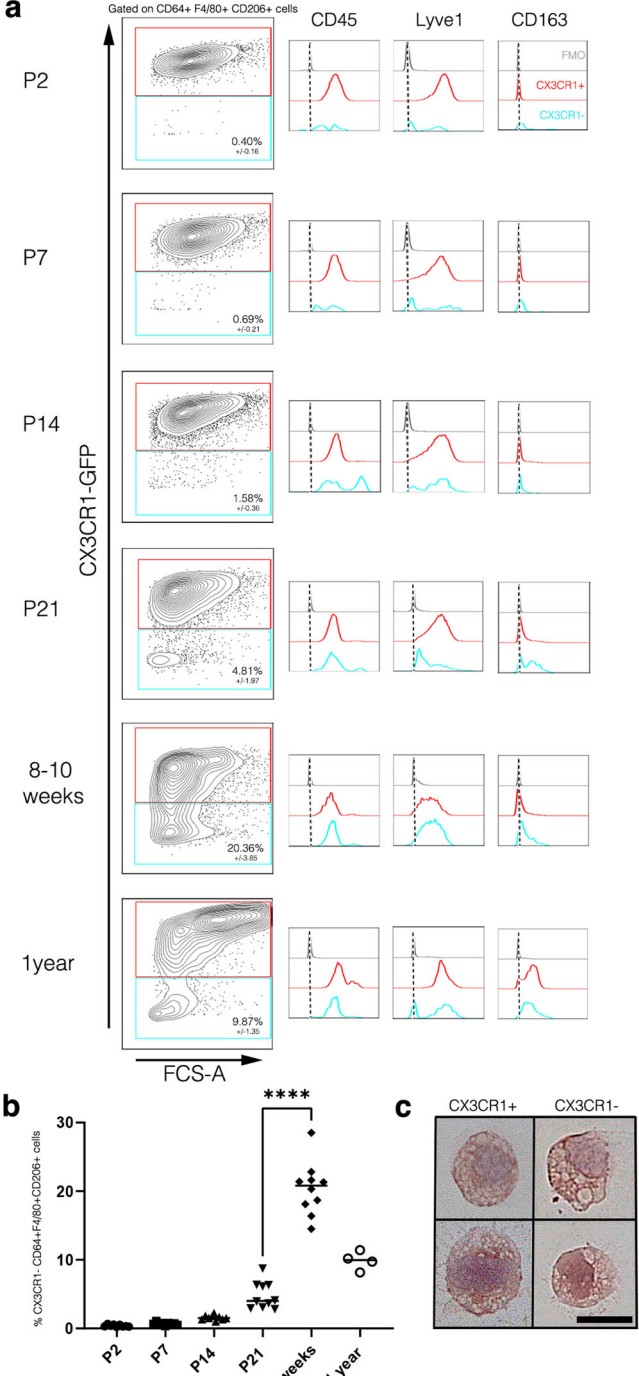

**Fig. 3 | Confirmation of the Lyve1⁺CX3CR1⁻ population by flow cytometry.** Flow cytometric analysis of *Cx3cr1ᴳᶠᴾ* brain parenchyma, pre-gated on living, single cells, CD64⁺, F4/80⁺, and CD206⁺ which excluded monocytes and neutrophils (gating strategy in Supplementary Fig. 3a). Microglia cells were excluded since these are Lyve1⁻CD11bʰⁱᵍʰ (Supplementary Fig. 3b). *Cx3cr1 ᴳᶠᴾ/⁺* mouse brains were analyzed at **a** P2 (3 individual experiments, *n* = 9 animals), P7 (3 individual experiments, *n* = 9 animals), P14 (3 individual experiments, *n* = 9 animals), P21 (3 individual experiments, *n* = 10 animals), adult (8–12 weeks after birth, 3 individual experiments, *n* = 10 animals) and 1-year old brains (1 individual experiment, *n* = 4 animals) showing CX3CR1⁺/CX3CR1⁻ gating for CD64⁺F4/80⁺CD206⁺ cells and subsequently on histograms for CD45, Lyve1, and CD163 expression in blue for CX3CR1⁻ and in red for the CX3CR1⁺ population. FMOs are shown in gray. **b** Total number of CX3CR1⁻F4/80⁺CD64⁺CD206⁺ cells at the different stages. **** represent *P*-value ≤0.0001 (unpaired *t*-test). **c** Morphological characteristics of cytospins from CX3CR1⁺ and CX3CR1⁻ pvMs sorted from brain and analyzed by hematoxylin/eosin staining. Two representative cells are shown for each population. Scale bars represents 10 μm (1 individual experiment, *n* = 5 animals). Source data are provided as a Source Data file.

stages were integrated and did not show changes during aging (data not shown and ref. 37). We focused on the pvM population and identified a single pvM cluster which expressed *Lyve1*⁺ (Fig. 4b, c, violin plots of all clusters in Supplementary Fig. 4a). This cluster also expressed the macrophage markers *Mrc1* (encoding CD206), *Emr1* (encoding F4/80), *Fcgr3* (encoding CD64), and others (Fig. 4c). However, this pvM cluster did not express *H2-Aa*, the genes encoding MHCII in C57Bl/6 mice (Fig. 4b, c). Generally, *Cx3cr1* and *Ptprc* expression levels were lower than in other pvM/BAM populations or in the microglia cluster which was identified by high *Siglech* expression (Fig. 4b, c). Isolating the cells from the pvM cluster with the highest *Lyve1* expression revealed no further unsupervised segregation based on *Cx3cr1* or *Ptprc* (Fig. 4e). However, there appeared to be 2 segregated pvM subclusters, based on e.g., *Lyve1*, *CD209f*, *CD209g* expression in subcluster 1 and *Atf3*, *Fos*, *Junb* in the other (Fig. 4d and Supplementary Fig. 4b).

Taken together, our data rule out a lymphatic identity of the CX3CR1⁻Lyve1⁺ cells and establish that these cells are bona-fide pvMs.

### The origin of CX3CR1⁻ Lyve1⁺ perivascular macrophages

PU.1 is considered to be a master regulator of macrophage differentiation. Therefore, we analyzed the *Spi1ᴳᶠᴾ/ᴳᶠᴾ* mice, in which the *Spi1* locus (encoding PU.1) is inactive and lack all macrophages and die around birth[9,38]. In E18.5 *Spi1ᴳᶠᴾ/⁺* embryos, we identified Lyve1⁺-PU.1⁺CD45⁺ and Lyve1⁺PU.1⁻CD45ˡᵒʷ pvMs (Fig. 5a). In contrast, E18.5 *Spi1ᴳᶠᴾ/ᴳᶠᴾ* embryos lacked Lyve1⁺ cells in the brain parenchyma (Fig. 5b, positive control of Lyve1⁺ cells in the skull in Supplementary Fig. 5a). This demonstrated that Lyve1⁺CD45ˡᵒʷ pvMs depend on PU.1 during their development, but some cells downregulated its expression (Fig. 5c). All Lyve1⁺ cells within the sections also expressed F4/80 and CD206 (Supplementary Fig. 5b). To confirm lack of PU.1 expression, we stained the PU1-GFP with an anti-GFP and confirmed lack of GFP signal (Supplementary Fig. 5c).

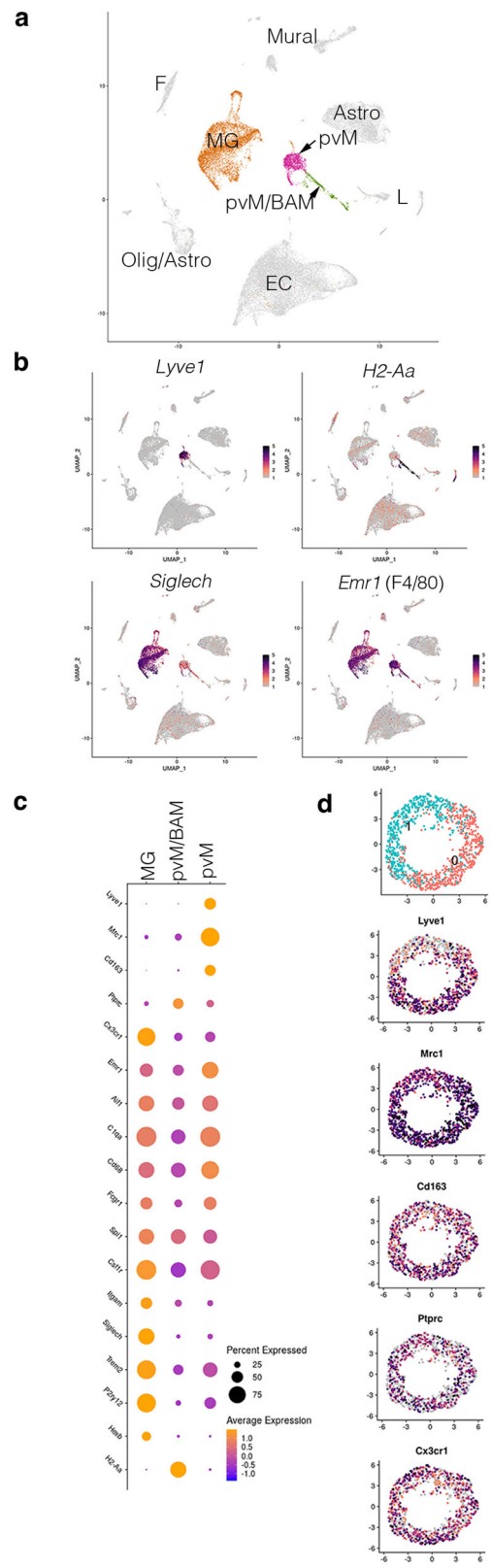

**Fig. 4 | Transcriptional profiling of the pvM subsets.** Replicates of mouse brain cortex, excluding neurons and meninges. Shown is the integrated dataset of samples at 10 days (*n* = 3, 12,843 cells), adult (7–11 weeks, *n* = 2, 7817 cells), and aged (1.5 years, *n* = 3, 10,448 cells). Cluster analysis on integrated data (**a**) focused on the pvM, the pvM/BAM and the microglia clusters. **b** The pvM cluster was indicated as the sole cluster with *Lyve1* expression, but *H2-Aa* (encoding one of the MHC-II genes) negative. This population was segregated completely from the *Siglech* expression microglia, which was also *Lyve1⁻*. *Emr1* (encoding F4/80) is expressed in all 3 clusters. **c** Dot-plot of the genes associated with macrophage or microglial identity, showing that pvMs is the only cluster which is expressing *Lyve1*. pvM/BAM is characterized by *H2-Aa* expression, but lacking *Lyve1*. The microglia population in cluster 3 was identified by lack of *Lyve1*, *CD163*, *H2-Aa*, and low *Mrc1* (encoding CD206) or *Ptprc* (encoding CD45) expression. EC endothelial cell, MG microglia, pvM perivascular macrophage, BAM border-associated macrophage, L Lymphocytes, F fibroblast, Astro astrocyte, Olig Oligodendrocyte, Mural Mural cells. **d** Cell from the cluster with the highest *Lyve1* expression, the pvM, were isolated to analyze a possible segregation in *Cx3cr1⁺* and *Cx3cr1⁻* expression population within this population. Even though we observed 2 subclusters, also based on *Lyve1* expression, there was no segregation based on *Cx3cr1* or *Ptprc*.

To establish whether the cells were derived from a CX3CR1 macrophage precursor, we crossed the reporter *Cx3cr1ᴳᶠᵖ* animals with mice harboring an inducible *Cx3cr1-Cre;Rosa26ᵗᵈᵀ* reporter. In the resulting animals tdTomato is expressed in all cells, which at some point in their ontogeny expressed CX3CR1, while GFP is expressed only in those cells which express CX3CR1 at the time of analysis. As expected, microglia expressed both tdTomato and GFP (Fig. 5d). All Lyve1⁺ pvMs expressed tdT but not all pvMs expressed GFP (Fig. 5d plus multiple sections in Supplementary Fig. 5d). Therefore, as for PU.1, the CX3CR1⁻ pvMs express CX3CR1 during their ontogeny, but subsequently lose this expression, as has been reported for other tissue macrophages including Kupffer cells[39].

The *Cxcr4-CreErt2* lineage-tracing model is used to label all embryo derived hematopoietic progenitors, but excludes yolk-sac-derived hematopoietic cells such as microglia and pvMs[10,27,28,40]. In the brains of naïve *Cxcr4-CreErt2;Rosa26ᵗᵈᵀ* mice injected with 4-OHT four weeks before isolation, to label all bone marrow hematopoietic stem cells (HSC), we observed that an average of 97.3% of the CD64⁺F4/80⁺CD206⁺CX3CR1⁺ pvMs and 96.7% of the CD64⁺F4/80⁺CD206⁺CX3CR1⁻ pvMs were tdTomato⁻ (Fig. 5e, gating strategy in Supplementary Fig. 5e). As validation for the model, and as was previously shown[40], microglia were not labeled in this model, while monocytes and neutrophils were labeled (Supplementary Fig. 5e). Collectively, these data establish that as previously suggested[10], in naïve animals CX3CR1⁻ and CX3CR1⁺Lyve1⁺ parenchymal pvMs are not derived from HSC, but their ontogeny involves a CX3CR1⁺ macrophage precursor.

## Functional analysis brain pvM populations
Besides their role as source for chemokines and growth factors to regulate immune responses, pvMs display phagocytic activity, which could be part of a broader role of these cells in tissue homeostasis[23,41]. To address a potential role of the CX3CR1⁻Lyve1⁺ pvMs in fluid drainage or macromolecule clearance, we injected 2.5 μl 10kD Dextran-AlexaFluor647 or Acetylated-LDL-AlexaFluor594 into the lateral ventricle of *Cx3cr1ᴳᶠᵖ* mice. To exclude recirculation through the blood stream, we analyzed the brain 10 min after injection[3]. We observed that both CX3CR1⁻ and CX3CR1⁺Lyve1⁺cells phagocytosed the injected dyes near the ventricle, in the dorsal and in the ventral cortex (Fig. 6a–f). We conclude that like their CX3CR1⁺ counterparts, CX3CR1⁻ Lyve1⁺ cells were able to efficiently take up lipoproteins and glycoproteins in the range of at least 3–10 kDa similar to conventional pvMs. The close proximity of the pia mater extending towards the third ventricle (3 V) could explain direct drainage from the injection-site within the lateral and connected 3 V towards the pia mater (Fig. 6g). However, pia dura drainage is unlikely to occur via LECs, since very few Lyve1 and Prox1mOrange2 positive nuclei were observed within the pia mater (Fig. 6g).

## Involvement in central nervous system diseases
Emerging data suggested that in stroke pvMs can influence blood–brain barrier function[42,43]. In patients with intracerebral hemorrhage and focal cerebral ischemia, an accumulation of CD163⁺

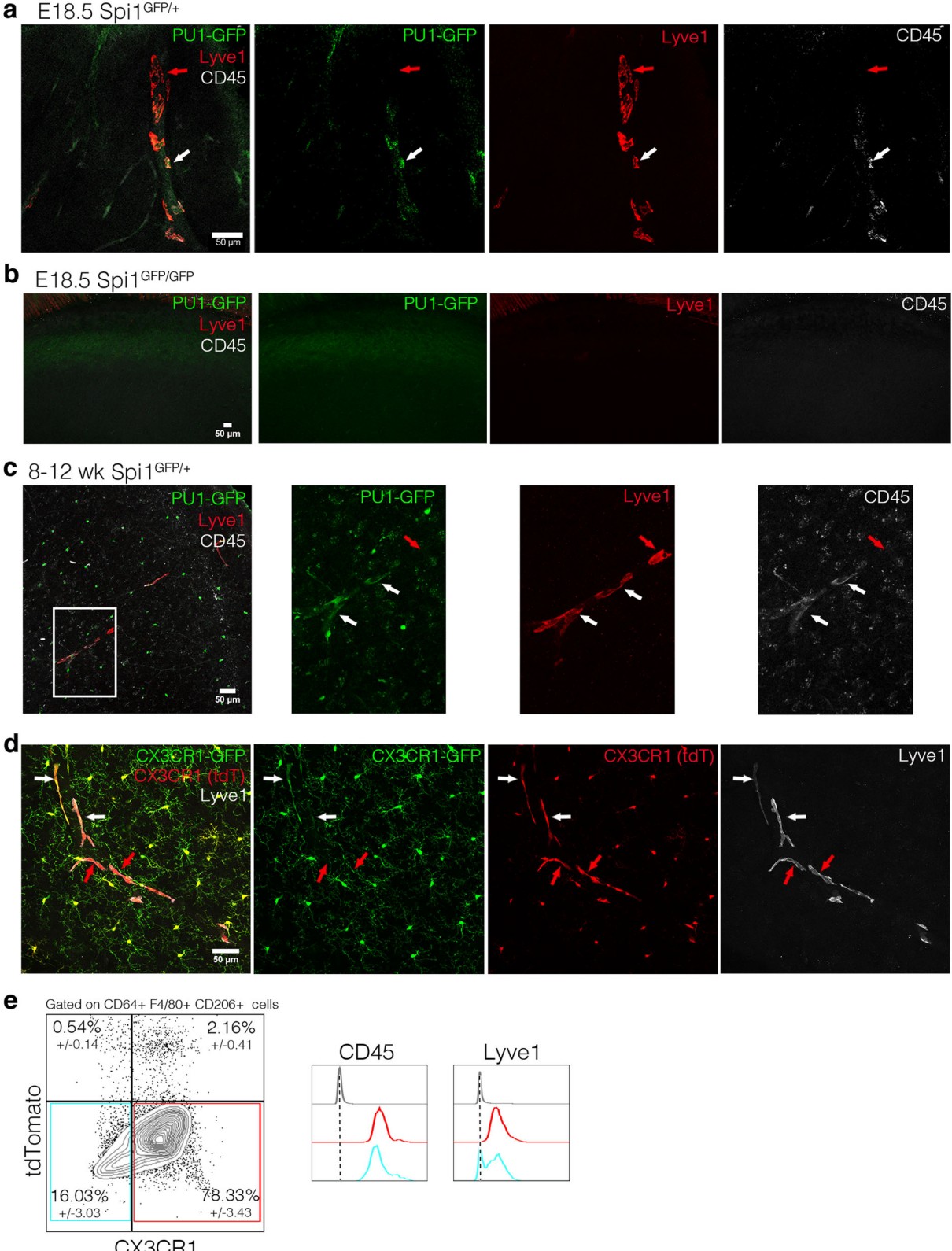

**Fig. 5 | The origin of CX3CR1⁻ Lyve1⁺ perivascular macrophages.**
**a** Immunofluorescence on sections of the head of a E18.5 *Spi1*^GFP/+ embryo (PU1-GFP in green), labeled for Lyve1 (red) and CD45 (white) (60 μm maximum intensity projection) shows the presence of the conventional (white arrows) and the non-conventional pvM populations (red arrows) at this stage (*n* = 3). **b** No GFP nor Lyve1 fluorescence was observed in the *Spi1*^GFP/GFP E18.5 dorsal cortex (33 μm maximum intensity projection) (*n* = 2). **c** Immunofluorescence on *Spi1*^GFP/+ adult mouse brain (48 μm maximum intensity projection) (*n* = 3). **d** Immunofluorescence on sections

of *Cx3cr1*^GFP/+; *Cx3cr1-Cre*; *Rosa26*^tdT mouse brain showing that conventional (white arrows) and non-conventional pvM population (red arrows) are expressing tdTomato (*n* = 3). **e** Flow cytometry analysis of *Cxcr4-CreErt2*; *Rosa26*^tdT brain parenchyma, gated on living, single cells and CD64⁺ F4/80⁺ CD206⁺ cells. Subsequently histograms of CD45 and Lyve1 expression of the CX3CR1⁻tdTomato⁻ population in blue, the CX3CR1⁺tdTomato⁻ population in red and FMO in gray (*n* = 3). Source data are provided as a Source Data file.

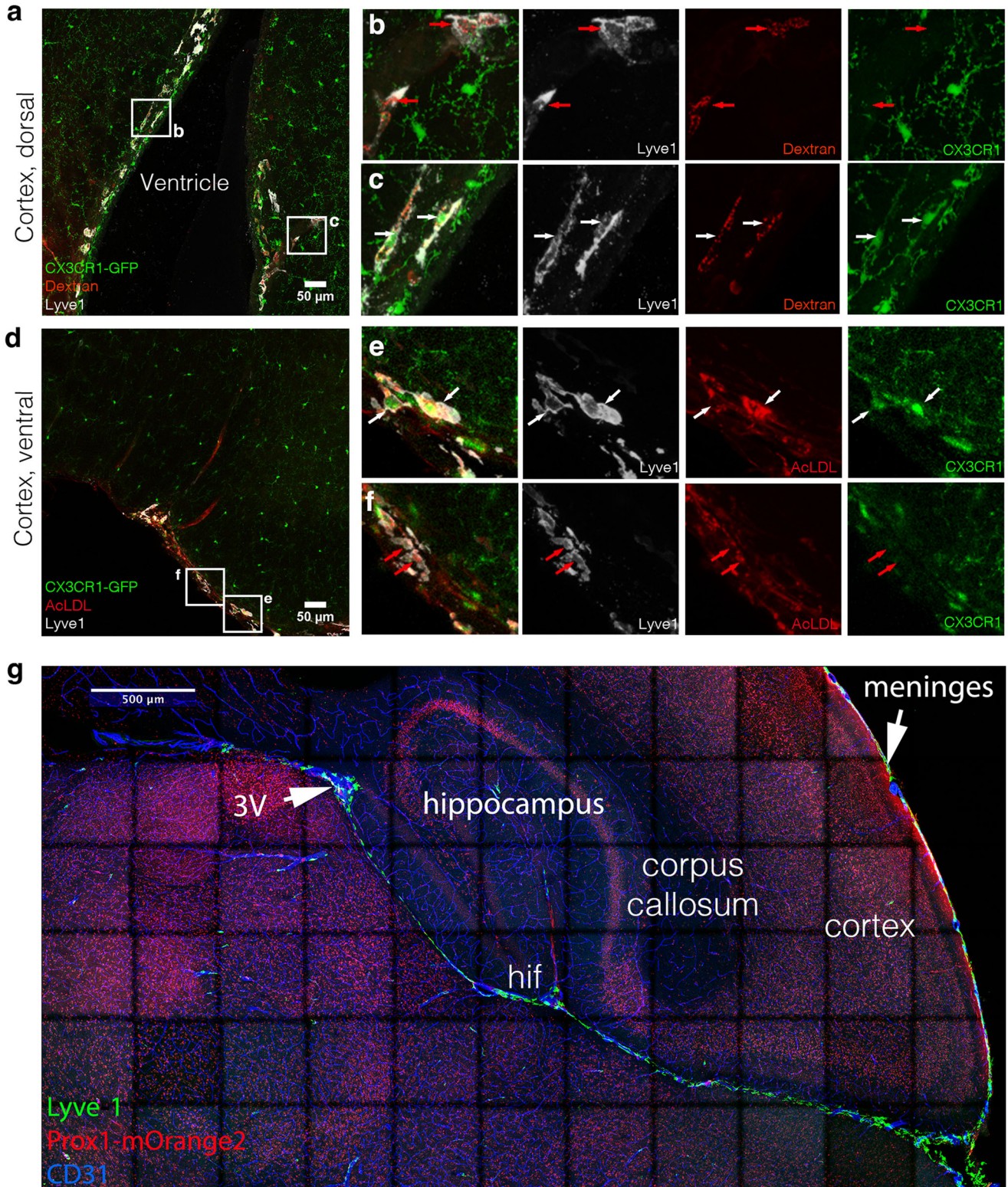

**Fig. 6 | CX3CR1⁻ pvM share phagocytic functioning with CX3CR1⁺ pvM.**
**a**–**c** *Cx3cr1^GFP* mouse brains were injected intra-ventricular with Dextran-Alexa647 (37 μm maximum intensity projection) (*n* = 3) and **d**–**f** Acetylated-LDL-Alexa594 (42 μm maximum intensity projection) (*n* = 3). Ten minutes after injection, mice were euthanized and brains dissected. Confocal analysis on brain sections shows Lyve1 (white), Dyes (red), and CX3CR1 (green). The unconventional pvMs phagocytosed the dyes (red arrows), as well as did conventional pvMs (white arrows) (**b**, **c**, **e**, **f**). **g** Tiled confocal acquisition of a *Prox1^mOrange2* brain section. Lyve1⁺ cells in green lined blood vessels stained for CD31 in blue. Lyve1⁺ cells in green penetrated the brain to the hippocampal fissure (hif) and all the way towards the third ventricle (3 V). (*n* = 4).

cells around brain blood vessels, which also contained myelin, was observed[44]. However, it remains unclear whether these cells are pvMs, other BAM or monocytes[45].

To assess the role of the pvMs in cerebrovascular pathologies, we investigated contributions of these cells in a photothrombosis (PT) model of ischemic stroke. PT was induced in 8-week-old male mice and we observed an increase in Lyve1 staining density in whole-mount stained brains (Fig. 7a, quantification of whole-mount staining density as shown in Supplementary Fig. 6, Supplementary Movies 5 and 6). At day 14 after stroke induction (P14), we noticed a significant increase in Lyve1 staining density within the hippocampus (4.2x, $p < 0.05$), although no significant difference was observed in the dorsal cortex (Fig. 7b). At P30, Lyve1 staining density had normalized to control values (Fig. 7b). We confirmed the increase in the pvMs by flow cytometry at P3, P8, and P14 after stroke using $Cx3cr1^{GFP}$ mice (Fig. 7c, d). After PT, the CX3CR1$^+$Lyve1$^+$F4/80$^+$population increased, while the CX3CR1$^-$Lyve1$^+$F4/80$^+$ population decreased (Fig. 7c, d).

Such an increase in pvMs could be due to the migration of bone-marrow-derived monocytes into the lesion and subsequent to differentiation to pvMs, or result from local proliferation of the tissue resident pvMs. Lyve1$^+$ cells in and near the lesion were not positive for Ki67 immunofluorescence (data not shown). To determine if the increase originated from bone marrow HSC, we analyzed Lyve1$^+$F4/80$^+$ pvMs in the brain of $Cxcr4$-$CreErt2$;$Rosa26^{tdT}$ male mice, which received 4-OHT four weeks before PT induction[40]. As observed under naïve conditions (Fig. 5e), most pvMs were tdTomato$^-$, and there was no increase in tdTomato positive cells at 14 days after PT induction (Fig. 7e), suggesting local (progenitor) proliferation of pvMs rather than replacement by bone-marrow-derived monocytes.

## Discussion

Here we provide an in-depth study of the perivascular macrophage compartment at the brain borders, including high end imaging, fate mapping as well as phenotypic and transcriptomic profiling.

Detailed analysis for the presence of lymphatic endothelial cells within the CNS using different reporter mice combined with immunostaining for lymphatic endothelial cell markers did not reveal evidence for these cells within the parenchyma. Only few isolated lymphatic endothelial cells were observed within the pial meninges, as described previously[35]. Using flow cytometry, single-cell RNA sequencing, immunofluorescence and lineage-tracing models, we observed that the pvM population within the CNS was almost exclusively Lyve1$^+$, CD45$^{low-int}$ and lacked MHC-II expression. Low MHC-II expression on peripheral perivascular Lyve1$^{hi}$ macrophages was described before[23]. How the Lyve1$^+$MHC-II$^-$ brain parenchymal pvMs relate to these peripheral Lyve1$^{hi}$MHC-II$^-$ monocytes remains unclear. We observed very few Lyve1$^-$MHC-II$^+$ cells within the parenchyma, and the single-cell RNA sequencing clearly separated the Lyve1$^+$MHC-II$^-$ and Lyve1$^-$MHC-II$^+$ pvM subsets. However, we cannot rule out contamination with choroid plexus cells containing BAMs, which might explain the lower CD163 and higher CD45/$Ptprc$ expression of these cells. The Lyve1$^+$ pvMs were located within the perivascular space around arterioles mainly, but not veins. This confirms a recent study in which Lyve1$^+$ pvMs were detected around Cx40 positive arteries and large EphB4 positive veins[25].

Our data show that the brain parenchymal pvM population is more heterogenous than previously described and include a non-conventional CX3CR1$^-$Lyve1$^+$ subpopulation. These expressed classical macrophage markers such as F4/80, CD11b, CD206, CD64, and CD163 and clustered together with CX3CR1$^-$Lyve1$^+$ pvMs in single-cell sequencing analyses. The CX3CR1$^-$Lyve1$^+$ population peaked at adulthood reaching around 20% of the Lyve1$^+$ pvM population and its numbers decreased in aged animals. Single-cell RNA sequencing data however did not reveal differences in the gene expression patterns between these stages, indicating that the population did not change drastically over time. Similar to their CX3CR1$^+$ counterparts, the

CX3CR1$^-$Lyve1$^+$pvMs were situated outside the cul-de-sac of the pia mater and within the Virchow-Robin space, which is likely to be associated with a function in drainage or clearance of brain parenchyma-derived metabolites or breakdown products[9,43,46]. In adult tissues at steady-state, pvMs are believed to have important functions related to their perivascular location, such as the regulation of vascular permeability, phagocytosis of blood-transmissible pathogens, antigen presentation, or immune regulation[9,41]. Analysis of phagocytosis activity, revealed that CX3CR1$^-$ pvMs displayed similar uptake as CX3CR1$^+$ pvMs.

Fate-mapping studies have established that many tissue macrophages, such as alveolar macrophages, Langerhans' cells and Kupffer cells originated from CX3CR1$^+$ precursors but cease to express $Cx3cr1^{37}$. In a recent detailed study on the origin and lineage of the pvM, it was shown that the pvMs are derived from the yolk sac hemogenic endothelium and depend on local factors to differentiate[10]. The use of the Spi1$^{GFP/GFP}$ (PU.1) and $Cx3cr1$-$Cre$ models showed that the CX3CR1$^-$ pvMs are most likely derived from the same lineage as the pvM population, thus, from yolk-sac-derived progenitors. Furthermore, our data suggest that replacement and increase of CX3CR1$^-$Lyve1$^+$ cells is independent from monocytes by using the inducible $Cxcr4$-$CreErt2^{tdT}$ lineage-tracing model[40], as was shown in the naïve situation recently[10]. We extend these data, by establishing that also the pvMs increase after stroke is mostly due to local proliferation with limited monocyte contribution.

In conclusion, here we demonstrate that the majority of brain parenchymal pvMs are Lyve1$^+$MHCII$^-$CD45$^{low/int}$ and the population is more heterogenous than previously described. The specific function of non-conventional CX3CR1$^-$ pvMs as compared to the conventional pvMs remains to be determined. These cells are consistently observed within the brain and may act in drainage or clearance of brain parenchyma-derived metabolites or breakdown products which have been proposed to flow along arterioles into the CSF[11]. Presently, there are no unique markers available to target these cells specifically, and to study their specific role in brain physiology in the future.

## Methods
### Mice
C57BL/6 J (stock number 000664) mice were obtained from Charles River (France). $Prox1$-$CreERT2^{+/-\ 34}$, $Wnt1$-$Cre^{39}$, and $Cx3cr1^{GFP}$;$Cx3cr1$-$Cre^{tdT}$ mice were kindly provided by Dr. Bajénoff (CIML, Marseille, France). $Cx3cr1^{GFP}$ mice[47] were kindly provided by Dr. Lelouard (CIML, Marseille, France). Spi1$^{GFP}$ (PU.1)[38], $Cxcr4$-$CreErt2$;$Rosa26^{tdT\ 40}$, and $Prox1^{mOrange2\ 48}$ were maintained at the CIML at SPF conditions (Marseille, France). All mice were co-housed in the same room under similar conditions with water and food ad libitum and 12 h/12 h night/daylight cycle.

$Prox$-$CreERT2^{+/-}$ and $Wnt1$-$Cre$ lines were crossed to homozygosity for the tdTomato reporter using Rosa26$^{tdT}$ mice[49]. 1.6 mg 4-hydroxy-tamoxifen (4OHT) was injected IP 4 weeks before the isolation of the $Cxcr4$-$CreErt2$;$Rosa26^{tdT}$ adult brain and 2 weeks before the isolation of the $Prox$-$CreERT2^{+/-}$;$Rosa26^{tdT}$ brain. All animal studies were reviewed and approved by the local ethics committee of Aix-Marseille University and the Ministère de l'Enseignement Supérieur, de la Recherche et de l'Innovation. We used males for the experiments.

Antibodies in Supplementary Table 1

### Cell preparation and flow cytometry
$Cx3cr1^{GFP}$ mice were euthanized with an overdose of ketamine-xylazine anesthetic and perfused with phosphate buffered saline (PBS) plus heparin. Brains were isolated, dura was removed, and the brain cut sagitally in 6–8 pieces, then digested with the Adult Brain Dissociation Kit (Miltenyi, 130-107-677) at 37 °C for 30 min on the gentle MACS dissociator (Miltenyi). Brain cell suspensions were filtered over 70 μm strainers and filters were washed with 10 mL HBSS 2%FBS. After centrifugation (5 min, 400 × $g$, 4 °C), debris were removed using a 40%

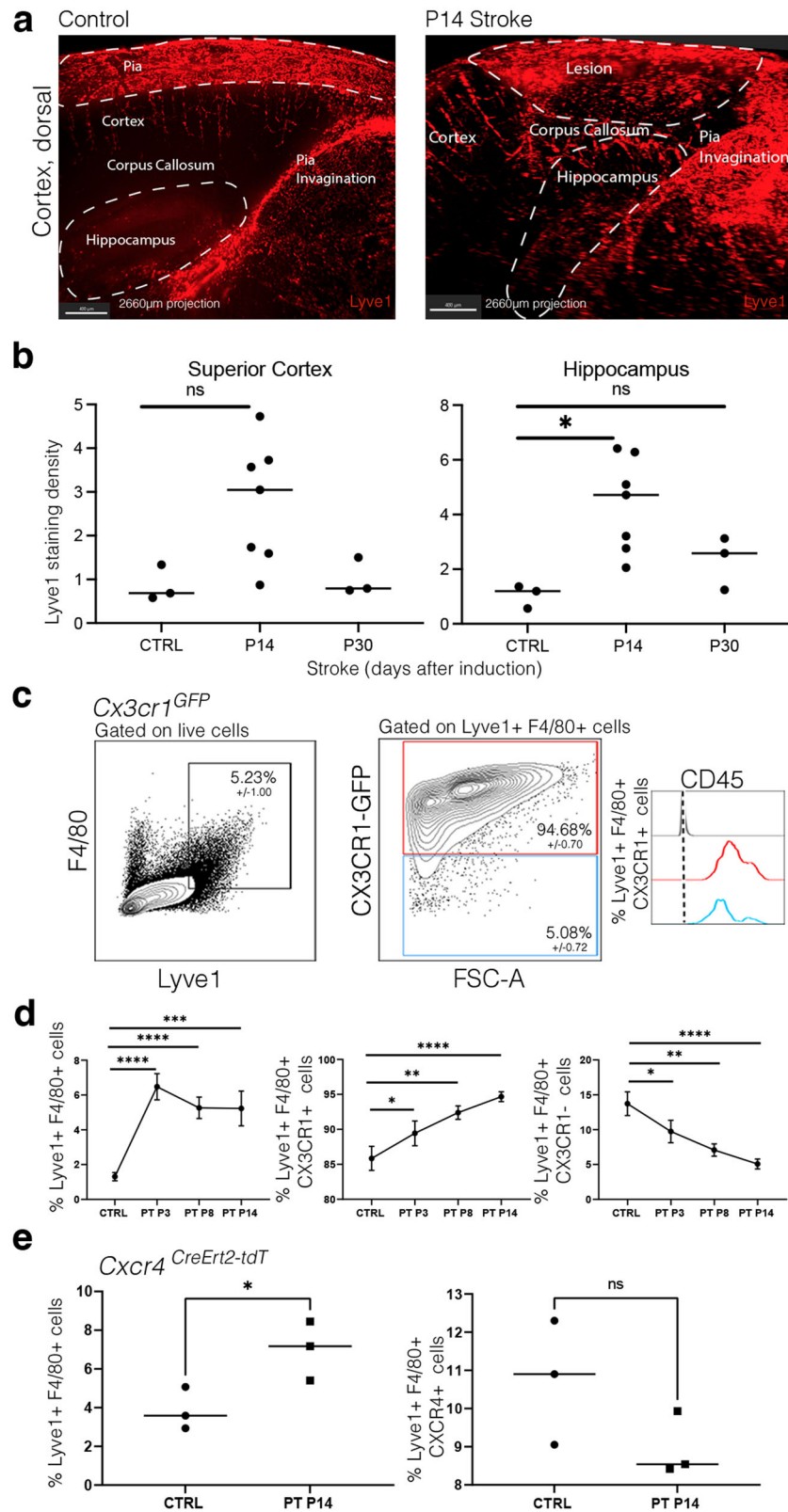

Percoll (Sigma–Aldrich, GE17-0891-02) solution in PBS. Cells were centrifuged (30 min, $500 \times g$, 4 °C) and supernatant was discarded. Red blood cell lysis was performed for 10 min at 4 °C using the Miltenyi RBC lysis buffer from the kit. Lysis was stopped with 9 mL HBSS 2%FBS and cells were centrifuged (5 min, $400 \times g$, 4 °C). Cells were then blocked (15% normal mouse serum (Jackson Immunoresearch 015-000-120) in FACS buffer (HBSS 2% FBS) for 15 min and subsequently

stained for CD45-BUV395 (BD biosciences 564279), Lyve1-eFluor660 (eBioscience 50-0443-82), F4/80-BV421 (Biolegend 123131), CD64-BV711 (Biolegend 139311), CD206-BV785 (Biolegend 141729), CX3CR1-PECy7 (Biolegend 149015), Ly6C-BV510 (Biolegend 128033), LY6G-AF700 (BD Pharmingen 561236), and CD163-SB600 (eBioscience 63-1631-82) diluted in FACS buffer for 30 min on ice. The staining of dead cells with Fixable NIR was performed for another 30 min after the

**Fig. 7 | CX3CR1⁺ pvM numbers increase after induced ischemic stroke.**
**a** Maximal intensity projections (sagittal view) (2660 μm) of lightsheet acquisitions on cleared brains, stained for Lyve1 (red), control and after stroke (P14 = 14 days after induction of ischemic stroke) (2 individual experiments, $n = 7$ animals) (Supplementary Movies 5 and 6). **b** Quantification of Lyve1 in the area containing the whole stroke lesion within the dorsal cortex situated just above the hippocampus, imaged by lightsheet microscopy. * represents $P$-value ≤0.05 (Ordinary one-way Anova test). **c** Representative flow cytometry plots from *Cx3cr1^GFP* brain parenchyma 14 days after induced stroke (P14), pre-gated on living, single and showing Lyve1⁺F4/80⁺ cells and CX3CR1⁺/CX3CR1⁻ gating for these cells Lyve1⁺ F4/80⁺ and subsequently in blue the CD45 expression of the Lyve1⁺F4/80⁺CX3CR1⁻, in red the

CD45 expression of the Lyve1⁺F4/80⁺CX3CR1⁺ and FMO in gray. Microglia cells were excluded since they are Lyve1⁻. Average percentages for the different subpopulations are shown in the plots ($n = 4$). **d** Graphs representing the Lyve1⁺F4/80⁺ cell % and Lyve1⁺F4/80⁺CX3CR1⁺ cell % showing an increase of these populations at P3, P8, and P14 after stroke by flow cytometry and Lyve1⁺F4/80⁺CX3CR1⁻ cell % showing a decrease of this population at P3, P8, and P14 after stroke. Data are presented as mean value ± SD. Asterisk (*) represents $P$-value ≤0.05, **$P$-value ≤0.01, ***$P$-value ≤0.001, and ****$P$-value ≤0.0001 (unpaired $t$-test). **e** Graphs representing the Lyve1⁺F4/80⁺ cell % and Lyve1⁺F4/80⁺CXCR4⁺ cell % showing an increase of the first population at P14 after stroke by flow cytometry. Asterisk (*) represents $P$-value ≤0.05 (unpaired $t$-test). Source data are provided as a Source Data file.

antibody staining in protein free HBSS right after washing the cells. Cells were subsequently washed and resuspended in FACS buffer. Samples were acquired on the LSRFortessa X-20 cytometer (BD Biosciences). Data analyses were done using FlowJo software (version 10, FlowJo, LLC).

### Cell sorting
Cells were prepared and stained with Lyve1 and F4/80 antibodies and Fixable NIR as describe above. Live Lyve1⁺ F4/80⁺ Cx3cr1⁻, live Lyve1⁺ F4/80⁺ Cx3cr1⁺ cells were sort-purified separately using the BD FACS ARIA III SORP sorter.

### Cell morphology
Sorted cells were spun onto Shandon Cytoslides with a Thermo Shandon Cytospin 4 cytofuge (4 min at $400 \times g$), fixed with methanol and stained with hematoxylin and eosin.

### Gene expression analysis
Cerebral cortices were dissected from brains of perfused 10-day, 8–10-week, and 50-week-old male C57/Bl/6 mice, minced into small pieces in ice cold DMEM (Sigma–Aldrich) and digested at 60 min at 37 °C in an enzyme-cocktail containing 30U/ml papain (LK003153; Worthington), 40 μg/ml DNase type IV (LS006331; Worthington), and 0.125 mg/ml Liberase TM (5401119001; Sigma–Aldrich) in DMEM. Myelin was removed by mixing the cell slurry with 22% BSA in PBS and centrifugation at $1000 \times g$ for 10 min; pellets were collected. Single-cell suspensions were prepared and cells were counted using Luna-II automated cell counter (Logos Biosystems). For scRNASeq, cells were encapsulated in Gel bead emulsion using the 10X Chromium system (10X Genomics). Libraries were prepared according to the manufacturer's instructions using Chromium Single Cell 3′ Library & Gel Bead Kit v3 (10X Genomics) and sequenced in Illumina NextSeq 500 using High Output Kit v2.5 (150 cycles, Illumina). The Cellranger (10x genomics) pipeline was followed to demultiplex the sequencing data, to align it to the mouse reference genome (mm10), to count the reads, and to aggregate the resulting cell-gene matrices according to sample replicates.

Gene expression data analysis was performed with Seurat R package[50] (version 4.1.0). Quality control was performed on each sample replicate from juvenile, adult and aged time-points to remove poor quality cells and genes from the analysis using Create-SeuratObject function. Genes expressed in less than three cells were removed from the analysis as well as cells expressing less than 200 genes. Cells were also removed using an in-house developed R script that filter cells based on their number of detected genes, number of UMIs and percentage of mitochondrial and ribosomal genes using values in the table below. Data from sample replicates from a same time point were merged together and their respective expression matrices were log-normalized using NormalizeData function with a scale factor of 10,000). Their respective variable genes were found by FindVariableFeatures function using the "vst" method and the top 2000 variable genes were selected. Merged datasets from the three time-points (juvenile, adult and aged) were integrated using genes

selected by SelectIntegrationFeatures with nfeatures = 2000 then using FindIntegrationAnchors and IntegrateData with default parameters. The first 30 principal components from Principal Component Analysis were used for the UMAP non-linear dimensionality reduction and cells were clustered using Louvain clustering from FindClusters function with a resolution of 0.4. The differentially expressed genes from each cell cluster were identified using the Wilcoxon Rank Sum method from FindAllMarkers function with minimum fraction of cells expressing the gene over 10% and with a log fold-change upper than 0.25. Cellular identity of each cluster was determined comparing those differentially expressed genes with known gene markers specific to cell type from previous studies (Supplementary Table 2).

### Whole-mount staining and lightsheet imaging
Anesthetized mice were perfused with PBS-Heparin 5U/ml (Sigma) and subsequently overnight (ON) immersion fixed in paraformaldehyde 4% (PFA-Electron Microscopy Science, ref 15714) in PBS. The iDISCO protocol was used for immunostaining[51], prior to which brains were dehydrated in increasing methanol (MetOH) concentrations diluted in PBS (20, 40, 60, 80, and twice 100%) for 30 min for each concentration at RT. Subsequently specimens were incubated ON in a dichloromethane (DCM, Sigma 270997)-MetOH mixture (2 vol DCM:1 vol MetOH) at RT. After two 10 min incubations in absolute MetOH, brains were bleached in 5% $H_2O_2$ in MetOH ON at 4 °C and subsequently rehydrated by a decreasing MetOH series (80, 60, 40, 20% in water), followed by PBS and two washes in PBS-Triton X100 (Tx) 0.2% for 1 h each. Bleached brains were permeabilized for 2.5 days at 37 °C (0.4%Tx, 20%DMSO, 2.3% Glycine in PBS) and subsequently blocked with [PTwH (PBS, Tween20, Heparin), 10% DMSO, 6% serum] for 4.5 days at 37 °C. Whole-mount stainings were performed by incubation with primary antibodies for 5 days at 37 °C and subsequently with Alexa-dye coupled secondary antibodies diluted in PTwH containing 3% serum for 5 days at 37 °C. Following each staining step, samples were extensively washed in PTwH (10 min, 15 min, 30 min, 1 h, 2 h, and ON at RT). Finally, the samples were again dehydrated by increasing MetOH concentrations diluted in water (20, 40, 60, 80, 2×100% and ON in absolute MetOH), for 1 h each step at RT and cleared in a MetOH/BABB mix (1:1) [BABB (benzyl alcohol and benzyl benzoate = 1:2) (Sigma 305197 and Fisher Scientific 10654752)] for 8 h at RT and finally placed in BABB ON at RT to complete clearing. All incubations were done under mild agitation. After clearing, brains were imaged using a LaVision Ultramicroscope II (Miltenyi Biotec). Stacks were captured with a step size of 5 μm at ×2.5 magnification using an optic zoom with a NA = 0.144. 3D reconstruction, cell counting and analysis of the sample image stacks were performed using IMARIS software (Version 9.1.0, Bitplane). For Lyve1 quantification, an area of interest was selected (Dorsal cortex, Hippocampus) using the program's surface function and then a new channel corresponding to the fluorescence to quantify was created in this region. Using this workflow, a quantification of volume (in μm³) of Lyve1 stained material is possible, which was further adapted for volumetric quantifications (Supplementary Fig. 6).

## Vibratome section immunofluorescence staining and confocal imaging

Anesthetized mice were perfused with PBS/heparin, brains dissected and fixed ON in 4% PFA in PBS at 4 °C and subsequently embedded in 1% low melting agarose for generation of 100 μm vibratome slices (Leica, VT1000S). Sections were blocked in EBT buffer (EBSS, 0.05% Tx, 1% BSA) containing 10% serum for 2 h at RT under agitation. Immunostainings were performed by incubation in primary antibodies for 48 h at 4 °C in EBT, 3% sera and subsequently with Alexa-fluorochrome coupled secondary antibodies diluted in EBT, 3% sera for 24 h at 4 °C. Following each staining step, samples were washed several times in PBS-Tx (0.05% Tx in PBS) and in PBS. Sections were finally cleared in Histodenz (Sigma D2158) medium for 48 h at RT and subsequently mounted in Histodenz medium. Confocal images were acquired at RT on a confocal microscope (LSM880, Zeiss, Germany), with a ×20/0.4 Plan-Apochromat objective and using laser lines at 405, 488, 561, and 633 nm for the excitation of AlexaFluor405/GFP/Alexa-Fluor555/AlexaFluor647, respectively. Fluorescence was recorded in individual channels acquired in a sequential mode using a highly sensitive 32-channel GaAsP detector. Channels were, respectively, detected using these detection bands: A405 (410–470 nm), A488 (490–540 nm), A555 (565–640 nm), A647 (640–690 nm). The pinhole was set to 1 airy unit. Z stack were acquired with an optical thickness defined for each image in figure legends, satisfying the Nyquist resolution criterion. Image processing (contrast enhancement, scale bars, etc.) was done with ImageJ (National Institutes of Health) without actions modifying image integrity.

## Infusion of tracers into lateral ventricle

Mice were anesthetized (150 mg/kg Ketamine and 10 mg/kg Xylazine) and fixed in a stereotaxic frame. The skull was thinned with a dental drill at a location of 0.95 mm lateral and 0.22 mm caudal from the bregma. A 30 G needle with a silica fiber tip (Phymed) was inserted into the right lateral ventricle 2.35 mm ventral to the skull surface as previously described[3]. Injection of 2.5 μL acetylated-LDL-Alexa594 (ThermoFisher) or Dextran-Alexa647 10 kDa (ThermoFisher) tracer was done at a speed of 0.5 μL/min using a high precision syringe pump. The needle was left in place for 5 min and slowly retracted confirming lack of detectable backflow. To avoid recirculation through the blood stream into the brain, mice were euthanized 10 min after injection[3]. The mice were perfused with PBS and 4%PFA/PBS, brains isolated and fixed ON in 4% PFA/PBS at 4 °C, subsequently washed and then stored at 4 °C in PBS until further analysis.

## Photothrombosis (PT) model of ischemic stroke

Stroke using the PT model was induced in C57BL/6 J, Cx3cr1^GFP, and Cxcr4-CreErt2;Rosa26^tdT mice. 1.6 mg of 4-OHT was injected 3 consecutive days in the Cxcr4-CreErt2;Rosa26^tdT mice 4 weeks before the induction of the lesion, based on a protocol previously used for induction of stroke within the Cxcr4-CreErt2;Rosa26^tdT model[40]. Mice were anesthetized by intraperitoneal Ketamine/Xylazine injection as before and the eyes of the mice were covered with Ocry-gel to protect them from light and dehydration. The skin on the skull was incised from the eyes to the neck and retracted to the edges of the skull. After retro-orbital injection of 100 μL Rose Bengal (Sigma, 330000) a Leica, KL 1600LED cold light source was placed in contact with the skull and the illuminated region was precisely adjusted (using a stereotaxic instrument) to 2.5 mm caudal of the Bregma and 2.0 mm to the Lambda. After illuminating a 1 mm diameter area at power 3 for 15 min, the skin was put back into place, stitched and the animals returned to their home cages. Food was provided in a plate and Buprenorphine (0.4 mg/ml) was added to the drinking water. Mice were euthanized between 14- and 30-days post-induction and brains were analyzed by whole-mount immunofluorescence and flow cytometry.

## Statistical analysis

Graphs, average values and standard deviation (SD) shown in all figures were calculated using Prism (version 8.3.0 GraphPad software) software. One-way Anova test was used to determine significance. The number of individual experiments can be found in the legends of all the figures. Photoshop software (CC2015, Adobe) was used to generate figures.

## Reporting summary

Further information on research design is available in the Nature Portfolio Reporting Summary linked to this article.

## Data availability

All datasets generated for this study are included in the article/Supplementary Material. Raw data (fastq files) and processed data (gene counts) for single-cell RNA-Seq analysis have been deposited in the Gene Expression Omnibus with the primary accession number, GSE133283[37]. Source data are provided with this paper.

## Code availability

Scripts and detailed instructions to reproduce the NGS analysis for this manuscript are available on GitHub repository https://github.com/JulieBvs/SPlab_PVM2.

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

## Acknowledgements

We acknowledge the PICSL imaging facility, notably Mathieu Fallet and Sebastian Mailfert, of the CIML (ImagImm), member of the national infrastructure France-BioImaging. Lionel Chasson of the histology platform for the relentless cutting. We thank Toufik Guelmami and Michel Pontier. The flow cytometry core facility, notably Marc Barad, Sylvain Bigot, and Laurence Borge. We thank Hugues Lelouard and Marc Bajénoff for providing us with mouse models. We thank Lucas Bertoia for his help and Sandrine Henri, Marc Bajenoff, and Guillaume Hoeffel for their discussions. We thank Steffen Jung for the discussions and support. The work was supported by the FRM Amorçage jeunes équipes (AJE20150633331), ANR ACHN (ANR-16-ACHN-0011), ANR PRCI (ANR-17-CE13-0029-01), A*midex Chaire d'excellence to SAvdP and institutional grants to the CIML from INSERM, CNRS, and Aix-Marseille University. The project leading to this publication has received funding from the German Research Foundation SO285/11-1, the Marie Sklodowska-Curie International Training Network ENTRAIN (Grant Agreement #813294), the « Investissements d'Avenir » French Government program managed by the French National Research Agency (ANR-16-CONV-0001) and from Excellence Initiative of Aix-Marseille University—A*MIDEX. France-BioImaging is supported by the French National Research Agency (ANR-

10-INBS-04). MHS was supported by institutional grants from TU Dresden, ANR-17-CE15-0007-01 and ANR-18-CE12-0019-03, INCa (13-10/405/AB-LC-HS), Fondation ARC pour la Recherche sur le Cancer (PGA1 RF20170205515), the European Research Council (ERC) under the European Union's Horizon 2020 research and innovation program (grant agreement number 695093 MacAge) and an Alexander von Humboldt Professorship at TU Dresden. R.H.A. and H.W.J. were supported by the Max Planck Society, the Leducq Foundation and the DFG programs SFB1348, SFB1009, SFB1366, and FOR2325. F.K. is funded by Deutsche Forschungsgemeinschaft (DFG, German Research Foundation) SFB1348/1—386797833 and SFB1450/1—431460824, the Cells-in-Motion Cluster of Excellence (EXC 1003—CiM, 194347757), and the IZKF Münster (Kief / 019 / 20).

## Author contributions

C.S., M.L., H.W.J., M.S., S.K., S.K.R.S., K.K., S.W., H.H., L.F., and A.T. performed experiments. S.A.v.d.P. conceived and supervised the study. C.S., M.L., J.B., M.S., and S.A.v.d.P. analyzed data. C.S., M.L., H.W.J., M.S., A.T., M.F., S.S., M.H.S., R.S., L.S., R.H.A., S.S.M., F.K., and S.A.v.d.P. contributed to discussion and wrote, illustrated, reviewed, or edited the manuscript. All authors approved the submitted version of the manuscript.

## Competing interests

The authors declare no competing interests.
