## [Peer Review File · Nature Communications]

Deciphering the heterogeneity of the Lyve1+ perivascular Macrophages in the mouse brainReviewers' comments:

Reviewer #1 (Remarks to the Author):

This manuscript describes a novel cell type that appears to be located near the vasculature of the brain and can phagocytose particles. Based on this and the F4/80 expression and Lyve-1 expression the authors conclude and entitle their paper a CD45-negative perivascular macrophage... The paper is interesting but I would make a few suggestions.

1) Based on the behaviour and the F4/80 data and the exclusion of other possible cell types the authors conclude these are macrophages. However the paper could have been entitled a non-conventional PROX-negative perivascular lymphatic cell... What I am saying is that the data are not convincing that these are macrophages. Lots of cells can take up molecules 10kd in size so this does not necessarily suggest they are macrophages. The F4/80 can be expressed on various immune cells so not necessarily macrophages. I am not sure you want to conclude these are macrophages.

2) Did these cells ever have CD45 the way they did have PU.1 which you missed by just looking for its expression. Did these cells then convert from a CD45+ to a CD45- population.

3) Similarly did they ever have VAV-1 or PROX-1. Without proper lineage tracing it is hard to know the origins of these cells.

4) Can you show that they can replicate after the stroke? Can they become BRDU or Ki67 positive?

5) In some of the panels in Figure 1 labeling the nuclei to convince us those slim strands are cells would be helpful.

Reviewer #2 (Remarks to the Author):

Siret et al. report on a previously unidentified population of non-neuronal cell associated with cerebral blood vessels and expressing the lymphatic vessel marker Lyve1 but lacking CD45, as well as other markers of brain-resident myeloid cells. Located near blood vessels, the cells resemble perivascular macrophages (PVM) but lack classical PVM markers (mrc1, CD163, etc.). The cells lack other lymphatic endothelial markers (Prox1), as well as markers of brain endothelial cells (CD31), astrocytes (GFAP, AQP4), fibroblasts (ER-TR7), or pericytes (PDGFRbeta). Tracing studies with the exchange factor Vav suggested that the cells are not hematopoietic or bone marrow derived, although require a classical hematopoietic transcription factor for their development, but not maintenance. In a single cell RNAseq this cell population clusters with brain macrophages. The cells can internalize dextran or LDL, suggesting phagocytic activity, and their number increases after focal ischemic injury. It is concluded that this population of cells represent a new class of PVM with potential involvement in brain diseases.

This paper suggests a new population of vascular associated cells in the brain, which is potentially important considering the increasing role of non-neuronal cells in brain diseases. However, there are a number of issues that remain to be addressed or require clarification.

1. The evidence that these cells are macrophages is not convincing. They lack all myeloid cell markers, and the only evidence is related to their clustering with brain macrophages which is not sufficient to make this statement with confidence.
2. Another problem is that the origin of the cells remains to be defined. Are they yolk sac derived, like some brain resident innate immune cells? A clear lineage has not been established, which complicates determining their identity.
3. The perivascular localization is also not clearly documented. What size vessels are the cells associated with? Mention is made that they are embedded into the endothelial basement membrane, like pericytes, which would not place them into the perivascular space.
4. The fact that the cells pick up dextran does not define them as phagocytic. Other evidence of classical phagocytosis needs to be provided (particle size dependence, phagosome, pH, respiratory burst, etc.).
5. The single cell data is not described in sufficient detail. How were the cells isolated? Quality assessment, number of biological replicates, how many cells, how many genes/cells, age of the animals, etc.
6. Figure 1: What does “superior” or “inferior” cortex identify? What region of the hippocampus is depicted?
7. Figure 2A: this figure also shows conventional PVM and does not provide information on the lyve1+CD45- cells. The morphological relationships with the vascular wall are not well characterized at this resolution.
8. Figure 5: the lyve1+ CD45- cells seem to be rare
9. Figure 7: the labeling is not clear: what is pia? It seems too thick for pia mater. How can it be just above the hippocampus? Is there no corpus callosum? The location of the ischemic lesion is confusing. Does the stroke extend into the hippocampus? Which markers beside lyve1 do the cells express after stroke?

Reviewer #3 (Remarks to the Author):

In the current study by Siret et al, the authors describe a perivascular macrophage population positive for Lyve1 but negative for common macrophage markers including non-parenchymal brain macrophages like Cx3cr1, CD206 and CD163. These non-conventional perivascular macrophages (pvM2) were associated to vessels. While they expressed F4/80, they were not positive for PU.1 but needed it for their differentiation. Using Vavtdt mice, pvM2s were not labelled and therefore not of hematopoietic origin.

This is an interesting study and the identification of an additional non-conventional macrophage population associated to the neurovascular unit would be very important for the field.

However, the conclusions are not supported by their data and there are several issues with the current manuscript that need to be addressed.

The evidence is not convincing that these Lyve1+ cells are indeed macrophages:

- The ontogeny part was only addressed with one model (Vav1Cre), suggesting that they are in fact non-hematopoietic cells.
- The RNA-seq. data appeared to be performed on CD45- and Mrc1-expressing cells and does not unequivocally reveal the pvM2 population
- While they show nice images, they mostly only show Lyve1+ cells in combination with

markers for other cell types. Lyve1 alone is not sufficient to identify macrophages or even distinguish between pvM1 and pvM2s

- their flow cytometry data is not of sufficient quality

In more detail:

- Throughout the study, no quantifications are shown for their images. They mostly show 1-2 cells within a region and draw their conclusions based on a couple of cells. No information is provided about how many brains/images were analysed or how many times an experiment was repeated. They would need to show a quantification for each staining and region.

- They suggest that two populations of Lyve1+ pvMs are present, the conventional (Cx3cr1+) and the non-conventional (Cx3cr1-) ones. What is the percentage of Lyve1+ cells among pvMs that are Cx3cr-? Most stainings only show Lyve1 and they would need additional markers to identify them as conventional pvMs or pvM2 (for example Cx3cr1, CD206, CD163 etc.). Furthermore, they would need to show that the Lyve1+ cells are indeed macrophages by using additional stainings (F4/80 or Iba1) (Fig. 1G, H, I, Fig. 2, Fig. 5, Fig. 6, Suppl. Fig. 1, Suppl. Fig. 2).

- Overall, it would be nice to include DAPI, at least in the higher magnification images.

- In Figure 1, the authors describe that Lyve+Cx3cr1(GFP)- Iba1+F4/80+ macrophages lack CD45 and CSF-1R. However, the expression of both these markers is difficult to assess by immunohistochemistry. For example microglia and also some non-parenchymal macrophages are typically CD45^{lo}, which cannot be detected by histology, as evidenced by Fig. 1G where microglia (CD45^{lo}) are not positive for CD45. Thus, the statement that pvM2 are CD45 negative cannot be made by this image.

Regarding CSF-1R, the authors show one Lyve1+ cell, which is negative for CSF-1R.

However, this one specific cell highlighted with a red arrow appears to be very weakly positive for Lyve1 and whether this cell is indeed a cell and a pvM is not clear. As mentioned above, an additional staining such as F4/80 or Iba1 would be helpful and also Dapi. Thus, according to their image, it cannot be claimed that Lyve+Cx3cr1- pvMs lack CSF-1R expression.

For their flow cytometry data:

- They claim that microglia were excluded as they are Lyve1-. However, it seems that microglia are still included (Fig. 3) and that the two populations expressing F4/80, GFP and intermediate levels of CD45 are microglia and non-parenchymal macrophages.

Thus, Lyve1 does not look like a convincing staining and it is therefore absolutely critical that they show an isotype for Lyve1 for the different cell types and ages.

Other markers should be used in addition to identify microglia and non-parenchymal macrophages (for example CD64), and monocytes/neutrophils should be excluded. They should gate first on total macrophages and then show Lyve1 expression and the other markers.

- In addition, 60-80% of the CD45-Lyve+ cells are F4/80 negative (Figure 3). This clearly indicates that most CD45-Lyve1+ cells are NOT macrophages and emphasizes again that Lyve1 alone does not suffice to classify these cells as macrophages in the immunohistochemistry analysis.

- Figure 3E-F, why would they gate through the middle of the F4/80 population? These are

clearly all F4/80+ cells.

scRNA-seq data:

- It is not clear which cells were sequenced. In the material methods section, they state that for the RNA-seq data, they extracted Mrc1-expressing cells for analysis. However, in Supplementary Figure 1D, they claim that the pvM2 do not express CD206, meaning that their pvM2 population would not be included. On the other hand, they also write that the sequencing was performed on non-neuronal CD45- cells. However, most parenchymal macrophages are CD45^{lo} and would therefore have been excluded. (as stated above, they do not convincingly demonstrate that pvM2 are indeed CD45-).

In Figure 4B, the two clusters are not annotated. What are they supposed to be? Conventional pvM and pvM2 (red)? Yet, the cells expressing Lyve1, also express Spil1 (whereas in their reporter strain, pvM2 did not express Spil1). Also, Cx3cr1 seems to be expressed ubiquitously. Overall, the RNA-sequencing data did not reveal a cluster/population that corresponds to pvM2s.

- A transcriptome analysis of pvM2 would be critical to show that they are indeed macrophages and what markers they express.

- Vav1 fate-mapping: they demonstrate that the Lyve1+ cells are dtomato-. Again, quantification needs to be added. How many of the non-conventional pvMs are dtTomato-, in comparison to microglia and the Lyve1+ conventional PVMs for example?

The fact that the 'pvM2' are of non-hematopoietic origin suggests that these are not macrophages.

An additional model would be important to demonstrate this.

What is their ontogeny?

- Can pvM2 be found around veins and arteries? Have they looked at capillaries where no perivascular space is observed?

- There is no information about how many images and how many brains were analyzed or how many times an experiment was repeated.

We thank the reviewers for their interest in our study, their constructive comments and suggestions to improve the manuscript. We have addressed all issues raised by the reviewers and substantially improved the manuscript. This includes:

- An improved flow-cytometry panel including more macrophage markers in order to clearly establish the identity of the cells as macrophages. Also, we have added a comparison with microglia and now show the cell morphology of both pvM populations.

- One of the main aims of this study was to redefine the Lyve1⁺ pvM population, the most abundant pvMs within the brain notably on sections. Using a combination of immunofluorescence, flow cytometry and single cell RNA sequencing, we now described this pvMs in unprecedented detail. We have included a more comprehensive single cell RNA sequence dataset, based on a wider, non-neuronal population.

Immunofluorescence staining on sections revealed lack of CD45 staining only on the Lyve1⁺CX3CR1⁻ pvMs, while the Lyve1⁺CX3CR1⁺ cells were CD45⁺. However, since it is known that the signal for CD45 on brain sections is notoriously low to undetectable, similar as was observed for microglia (Fig. 1h, Goldmann et al., Nat.Imm. 2016, PMID 27135602), we cannot base the statement of CD45 negativity on only immunofluorescence. Based on the improved cytometry panel, we abstain to name the Lyve1⁺CX3CR1⁻ pvM population CD45 negative. Thus, CD45 negativity became less important than the new characterization we distinguished. Therefore, we have removed the statements on CD45 negativity, including in the title. Even though there was clear difference in macrophage expression within the Lyve1⁺ pvM, such as CX3CR1 or PU.1 on flow cytometry and in immunofluorescence, we could not observe a segregation within the *Lyve1*⁺ expressing cluster on these marker genes in the single cell RNA sequencing. Based on our extensive analysis of the pvMs we are convinced that the Lyve1⁺ pvMs is a heterogenous population, with some lost CX3CR1, some lost PU.1 and some have lost CD45 expression, but not necessarily all together. We propose that there is rather a spectrum of Lyve1⁺ pvMs than a bipolar distinction based on 1 marker. Hence, we removed the pvM2 annotation and rather describe the (sub)population based on the expression pattern, such as Lyve1⁺CX3XR1⁻.

- We have added information on the origin of the Lyve1⁺CX3CR1⁻ pvMs using different mouse models. We have previously used the PU.1 (*Spi1*^{GFP/GFP}) model to show a dependence on PU.1. We have now added the *Cx3cr1*^{Cre};*Rosa26*^{tdTomato} model, which labels the cells which during their ontogeny expressed CX3CR1. This model was used before to show that CX3CR1⁻ phagocytes expressed it before in their ontogeny (Yona et al., Immunity 2013, PMID 23273845). Second, to show the origin of cells which replace the pvMs, and replacement during stroke, we now used the *Cxcr4*^{CreERT2};*Rosa26*^{tdTomato} model which only labels HSC derived immune cells, i.e. bone-marrow derived monocytes, but not tissue resident macrophages (Werner at al., Nat Neuro 2020, PMID 32042176). Using this model, we now show that the pvMs were not replaced by bone-marrow derived monocytes in the naïve situation (providing supportive results to the recent data by the Prinz group, Masuda et al., Nature 2022, PMID 35444273). Moreover, using the same model we observed that the increase of Lyve1⁺ pvMs in stroke was not due to an invasion and replacement by the bone-marrow derived monocytes.

- A re-analysis of the single cell RNA sequencing data. We would like to note that this dataset was generated by the authors from the Adams group (Max Planck Institute, Münster). This dataset is under revision with eLife (preprint in BioRxiv <https://doi.org/10.1101/2022.06.10.495613>; Figure 1 attached). While publication from the Adams' group describes all clusters in more detail, and focuses on the endothelial cells and the changes of the MHCII⁺ pvM/Border-associated macrophages (BAM) during EAE, our manuscript focuses only on the *Lyve1*⁺ pvM cluster. If the reviewers would like to obtain more details on the other clusters present within the dataset, they can access this preprint (see Figure 1). We now clearly show that there is one *Lyve1*⁺ cluster within the complete non-neuronal cell population. Moreover, we now supply a dot-plot of the cluster, compared to the other pvM/Border associated macrophages (BAM) and microglia cluster with a set of macrophage markers to identify this population as true macrophages.

- We have added higher resolution images of the pvM localization and nuclei within the perivascular space, thanks to a collaboration with perivascular space experts, the lab of Lydia Sorokin.

Fig. 1 from Jeong et al., Adams lab MPI. Under revision with eLife, BioRxiv preprint <https://doi.org/10.1101/2022.06.10.495613>; Figure 1 in this study provides more details on the different clusters within the single cell RNA sequencing dataset we have used in this study, including endothelial cell (EC), Microglia (Micro), Astrocytes (Astro), Fibroblast (Fibro) and Mural cells (Mural). We have shown the complete dataset as well in Fig. 4, but focus only on the *Lyve1*⁺ pvM

Please find below our responses to the reviewers comments point-by-point:

Reviewer #1 (Remarks to the Author):

This manuscript describes a novel cell type that appears to be located near the vasculature of the brain and can phagocytose particles. Based on this and the F4/80 expression and Lyve-1 expression the authors conclude and entitle their paper a CD45-negative perivascular macrophage... The paper is interesting but I would make a few suggestions.

1) Based on the behaviour and the F4/80 data and the exclusion of other possible cell types the authors conclude these are macrophages. However the paper could have been entitled a non-conventional PROX-negative perivascular lymphatic cell... What I am saying is that the data are not convincing that these are macrophages. Lots of cells can take up molecules 10kd in size so this does not necessarily suggest they are macrophages. The F4/80 can be expressed on various immune cells so not necessarily macrophages. I am not sure you want to conclude these are macrophages.

We concur with the reviewer and therefore provide additional data on the identity of these cells which now more firmly establishing them as macrophages. We have redone the flow-cytometry in Fig. 3, with more macrophage markers and we compared the *Lyve1*⁺ pvMs with

the microglial population (Supplementary Fig. 3b). Moreover, we have provided cytospin stainings of the cells to show the morphology of the cells analyzed in flow-cytometry (Fig. 3c). Also, we provide a more detailed analysis on macrophage makers of the *Lyve1*⁺ cluster in the RNA expression data. Based on this data, we can firmly exclude a lymphatic endothelial cell identity and the cells described are bona-fide macrophages.

Second, we have now included the analysis on *Cx3cr1*^{Cre};*Rosa26*^{tdTomato} reporter mouse brains and show that the *Lyve1*⁺*CX3CR1*⁺ or ⁻ cells had expressed this typical macrophage marker during their ontogeny (Fig.5d). Also, during their ontogeny, these cells rely on PU.1 since we observed that in the PU.1 KO (*Spi1*^{GFP/GFP}) there are no *Lyve1*⁺ cells, thus pvM, within the brain parenchyma (Fig 5a-c).

Therefore, it is extremely unlikely that the cells described in this manuscript are of another lineage than a macrophage.

We have explained the use of the different mouse models plus the use of antibody stainings against lymphatic endothelial cell markers in the text to prevent any misunderstanding to rule out lymphatic origin (P4/5 line 120-125 and in the discussion, p9 1240-242). LEC critically depend on *Prox1* and all LEC express *Prox1*, albeit some at low levels, as was shown by others (e.g. Srinivasan et al., *Genes&Dev* 2007, PMID 17908929). Besides antibody staining (Fig. 2j-m), we also used the *Prox1*^{mOrange2} and *Prox1*^{CreErt2};*Rosa26*^{tdTomato} reporter mouse models (Supplementary Fig. 2b-d). Even though the *Prox1*^{low} (like collecting lymphatic vessels) would not be visualized in the antibody staining, this model will show reporter activity in lymphatic endothelial cells (and some neurons).

2) Did these cells ever have CD45 the way they did have PU.1 which you missed by just looking for its expression. Did these cells then convert from a CD45+ to a CD45- population.

This could indeed be true; CD45 is not required for macrophage development but is important for the adhesion process (Roach et al., *CurrBiol* 1997, PMID 9197241). We cannot rule out that pvMs lose CD45, or for the same matter, *CX3CR1* expression after they been integrated within the perivascular space. The major discrepancy with earlier publications on CD45 expression in brain pvM, is that brain pvMs were supposed to be CD45^{high}. On the contrary, we show a CD45 low/intermediate expression, similar as microglia (Figs. 3a and Supplementary Fig. 3b). They seem to be CD45 negative in immunofluorescence staining, and CD45 low-intermediate in flow cytometry, hence we now refer to CD45 as low to intermediate. Accordingly, we have adjusted the title of the manuscript.

In addition, we now show that next to PU.1, also *CX3CR1* expression is lost during ontogeny. In *Cx3cr1*^{Cre};*Rosa26*^{tdTomato} adult brains, we observed that the *CX3CR1*⁻ pvMs were tdTomato positive, indicating that these cells expressed *Cx3cr1* before in their ontogeny (Fig. 5d and Supplementary Fig. 5c).

The single cell RNA sequencing data also showed few cells with CD45 expression within the *Lyve1*⁺ pvM cluster (Fig. 4c, violin plots in Supplementary Fig. 4a and Supplementary fig. 4b). However, the main point of our manuscript it to show that the pvM population within the brain is extremely heterogenous, and some pvMs lose, or express at low levels, typical markers which are used for selection of pvM.

3) Similarly did they ever have VAV-1 or PROX-1. Without proper lineage tracing it is hard to know the origins of these cells.

Our data are in agreement with the recent publication by the Prinz group (Masuda et al., Nature 2022, PMID 35444273), in which they have shown in detail the origin of the pvM population. Since this publication described the (yolk-sac) origin of the pvMs in detail, this answers the comment from the reviewer on the origin of the pvM. We have referred to this publication in the discussion on the origin of the pvMs (p9/10, l269-271). However, the Masuda et al. publication addressed the pvM origins, but they did not specify pvMs heterogeneity within the adult.

Indeed, in the previous version of the manuscript, we did not observe labeling of the Lyve1⁺CX3CR1⁻ pvMs in the *Vav1^{iCre}* model, indicating that they were not derived from bone-marrow derived monocytes, nor from hematopoietic progenitors from the embryonic hemogenic endothelium. However, we have replaced the data from the *Vav1^{iCre}* model by more specific models to show their origin. These include the *Cxcr4^{CreErt2};Rosa26^{tdTomato}* reporter mouse model to specify their replacement from the bone-marrow derived monocytes and the *Cx3cr1^{Cre};Rosa26^{tdTomato}* reporter mouse model, next to the already shown data on the *Spi1^{GFP}* reporter mouse model, to determine a common macrophage origin.

On replacement of the pvM, we used the same *Cxcr4^{CreErt2};Rosa26^{tdTomato}* mouse model (Werner et al., Nat Neuro 2020, PMID 32042176) as the Masuda et al. 2022 publication, and we provide supportive data showed that pvMs are not replaced by monocyte derived macrophages in naïve mice. Moreover, we now show that during photothrombotic stroke the pvMs are not replaced by monocyte derived macrophages using the same monocyte specific reporter mouse line.

We have used the *Prox1^{CreErt2};Rosa26^{tdT}* model and injecting the tamoxifen 2 weeks before isolation did not reveal any labeling of the pvM. Moreover, data from the *Cx3cr1^{Cre}* and *Spi1^{GFP}* models plus the different macrophage markers expressed on these pvMs shown by flow-cytometry and single cell RNA sequencing indicate very strongly these cells are macrophages and descend from a macrophage lineage.

4) Can you show that they can replicate after the stroke? Can they become BRDU or Ki67 positive?

We thank the reviewer for this question. Indeed, any increase in cell numbers implies that there is proliferation. Related to this is the question whether resident pvMs proliferate or is the increase due to infiltrating monocytes?

We stained sections of the brain containing the lesions for proliferation marker Ki67. We observed much proliferation in or near the lesion, but Lyve1⁺ cells were in general Ki67 negative (white arrows in Fig. 2a attached), indicating not much proliferation. Also, the single cell RNA sequencing data showed not many cells from the Lyve1⁺ cluster expressing proliferative genes (Fig. 2b-c attached). We have addressed this in the results, p8 l230-231. Therefore, instead, to address the origin of the increased population we used the *Cxcr4^{CreErt2}* model. This model was introduced to distinguish between these subsets and was used to identify invading monocytes within the stroke lesion (Werner et al., Nat Neuro 2020, PMID 32042176; Ydens et al., Nat Neuro 2020, PMID 32284604). Similar as the study from Werner et al., we have injected the *Cxcr4^{CreErt2};Rosa26^{tdTomato}* with tamoxifen 21 days before PT to analyze infiltrating monocytes vs tissue resident macrophages. Our data show that in naïve mouse brains, 96% of the Lyve1⁺ pvMs are not HSC /monocyte derived macrophages (Fig 7e). During stroke this number was 93%. Of note, many monocyte derived monocytes do invade the brain after stroke (Werner et al., Nat Neuro 2020, PMID 32042176). However, we focused on the Lyve1⁺ pvM, which are not replaced.

Figure 2: a) Proliferation marker Ki67 staining in 50µm section of the PT lesion area revealed few Lyve1⁺ Ki67⁺ cells (white arrows). b) The UMAP plots of the different ages which were analyzed in the single cell RNA sequencing. The cluster associated with the Lyve1 expression is cluster 6 in juvenile (arrow), 3, in the adult and 7 in the aged populations. c) Expression of 3 genes associated with cell proliferation revealed that a small population within the cluster 6 of the juvenile was positive (arrows).

5) In some of the panels in Figure 1 labeling the nuclei to convince us those slim strands are cells would be helpful.

The lab of Lydia Sorokin, an expert on laminins and extra cellular matrix, with recent publications on the peri-vascular space (e.g. Zhang et al., JEM 2020, PMID 32379272) has addressed this. They have contributed to the resubmitted manuscript by a high-resolution confocal microscopy analysis of the CX3CR1⁺ and ⁻ pvM, and have shown that both cells indeed contain nuclei (Fig. 2g and Supplementary videos 3&4).

Reviewer #2 (Remarks to the Author):

Siret et al. report on a previously unidentified population of non-neuronal cell associated with cerebral blood vessels and expressing the lymphatic vessel marker Lyve1 but lacking CD45, as well as other markers of brain-resident myeloid cells. Located near blood vessels, the cells resemble perivascular macrophages (PVM) but lack classical PVM markers (mrc1, CD163, etc.). The cells lack other lymphatic endothelial markers (Prox1), as well as markers of brain endothelial cells (CD31), astrocytes (GFAP, AQP4), fibroblasts (ER-TR7), or pericytes (PDGFRbeta). Tracing studies with the exchange factor Vav suggested that the cells are not hematopoietic or bone marrow derived, although require a classical hematopoietic transcription factor for their development, but not maintenance. In a single cell RNAseq this cell population clusters with brain macrophages. The cells can internalize dextran or LDL, suggesting phagocytic activity, and their number increases after focal ischemic injury.

It is concluded that this population of cells represent a new class of PVM with potential involvement in brain diseases.

This paper suggests a new population of vascular associated cells in the brain, which is potentially important considering the increasing role of non-neuronal cells in brain diseases. However, there are a number of issues that remain to be addressed or require clarification.

1. The evidence that these cells are macrophages is not convincing. They lack all myeloid cell markers, and the only evidence is related to their clustering with brain macrophages which is not sufficient to make this statement with confidence.

We have further substantiated the macrophage identity by redoing flow-cytometry including more macrophage and microglia markers. We setup a new panel to which we added Csf1R, CD64, CD206 and CD163 for cytometric analysis of adult Cx3cr1 brains and indeed showed that the Lyve1⁺CX3CR1⁻ cells are macrophages (Fig. 3a). Moreover, we compared the Lyve1⁺ pvM population to the microglia (Supplementary Fig. 3b). While CD45 expression in immunofluorescence on sections was not detectable on Lyve1⁺CX3CR1⁻ pvM, similar as for microglia, we observed a CD45 low to intermediate population in flow-cytometry, expressing CD45 at similar levels as microglia (Fig. 3a and Supplementary Fig 3b). We cannot exclude that these cells are not negative for CD45, so we now refer to CD45 as low to intermediate. Accordingly, we have adjusted the title of the manuscript. The major discrepancy with earlier publications on CD45 expression in brain pvMs is that the pvMs were previously described to express CD45 at high levels.

2. Another problem is that the origin of the cells remains to be defined. Are they yolk sac

derived, like some brain resident innate immune cells? A clear lineage has not been established, which complicates determining their identity.

We determined whether pvMs are replaced by monocyte derived macrophages during life and in stroke, using the *Cxcr4^{CreErt2};Rosa26^{tdTomato}* model. We have injected the *Cxcr4^{CreErt2};Rosa26^{tdTomato}* with tamoxifen 21 days before PT to analyze infiltrating monocytes vs tissue resident macrophages. Our data indeed fully support the data from Masuda et al that 89% of the Lyve1+ pvMs are not HSC /monocyte derived macrophages (Fig 5e), while we show additionally that in stroke the increased pvM cell percentages are not due to an invasion of and replacement by monocytes (Fig. 7e). The recent and detailed data on the yolk-sac origin by the group of Prinz (Masuda et al., Nature 2022, PMID 35444273) beautifully illustrated the origin of the initial pvM to which we refer in our manuscript.

3. The perivascular localization is also not clearly documented. What size vessels are the cells associated with? Mention is made that they are embedded into the endothelial basement membrane, like pericytes, which would not place them into the perivascular space.

We were careful not to mention that the cells were embedded, but only mentioned ‘closely associated’. The close and intricate association of the pvMs with the vessel could indeed give the impression that they are embedded. As observed by wholemount microscopy, and discussed in the manuscript (p9, l251-252), these vessels are mainly SMA1⁺ arterioles but not the venous capillaries. Of all venous blood vessels, only a large cerebral vein is associated with the Lyve1⁺ pvM. In collaboration with the lab of Lydia Sorokin, we have addressed their exact location. We now show high resolution figures of the Lyve1⁺CX3CR1⁺ and ⁻ pvMs within the peri-vascular space (Fig. 2g and Supplementary videos 3 & 4).

4. The fact that the cells pick up dextran does not define them as phagocytic. Other evidence of classical phagocytosis needs to be provided (particle size dependence, phagosome, pH, respiratory burst, etc.).

The phagocytosis experiment was meant to show extra evidence for the Lyve1⁺ pvMs to be involved in drainage, for which we used 2 (Dextran 10 kD and Acetylated LDL) different compounds previously established as macrophage phagocytic molecules.

5. The single cell data is not described in sufficient detail. How were the cells isolated? Quality assessment, number of biological replicates, how many cells, how many genes/cells, age of the animals, etc.

We have addressed this issue in detail in the re-submitted version. We now show the complete dataset of the non-neuronal cortex cell isolation. Summarized, the dataset is an integration of non-neuronal cortical brain cell suspension isolated at 3 time points; 10 days after birth, 10 weeks and 1.5 years of age with at least n=2 for each time point. The dataset is also used in another manuscript, in which more in-depth analysis on the endothelial cells and the pvMs are shown. If required, more information on these clusters can be found in the preprint here: <https://doi.org/10.1101/2022.06.10.495613>.

6. Figure 1: What does “superior” or “inferior” cortex identify? What region of the hippocampus is depicted?

We have changed this to the cortex, dorsal and cortex, ventral for superior and inferior cortex. However, we did not further specify which cortex region the pictures were made, as the staining observed did not differ within these regions. The region where the pvM staining was

most prominent was within the hippocampal fissure (HIF) region, also shown in Fig. 6g. We have added to the legend of Fig. 1 that the pvMs were localized within the HIF.

7. Figure 2A: this figure also shows conventional PVM and does not provide information on the lyve1+CD45- cells. The morphological relationships with the vascular wall are not well characterized at this resolution.

We have addressed this issue in the collaboration with the lab of Lydia Sorokin, and the high-resolution image of both a Lyve1⁺CX3CR1⁺ and ⁻ pvMs within the peri-vascular space (Fig. 2g and Supplementary videos 3 & 4).

8. Figure 5: the lyve1+ CD45- cells seem to be rare

Indeed, in general Lyve1⁺ pvMs are rare within the parenchyma as can be observed in the whole-mount videos. They require immuno-fluorescence on thick sections or whole-mount in order to observe them. On the other hand, cytometry results could be biased as macrophages are sticky and are more difficult to obtain in suspension. This is also why multiple approaches are required to trustworthily determine their phenotype, as each technique has its own advantage and disadvantage. We observed this population in all approaches, thus making it very likely we observed a true population. We have omitted the CD45 negativity exactly because of this reason; In immuno-fluorescence on sections the Lyve1⁺CX3CR1⁻ cells were almost exclusively observed without CD45 expression, while the Lyve1⁺CX3CR1⁺ pvMs were observed with (faint) CD45 expression on sections. However, flow-cytometry revealed that regardless of the CX3CR1 expression, all Lyve1⁺ pvMs had low to intermediate expression of CD45, similar to microglia (Supplementary Fig.2b). Therefore, due to this discrepancy, we cannot rule out that the staining on sections is not optimal and therefore rather make the point that the Lyve1⁺ pvMs are CD45 low to intermediate, which is different compared to earlier publications where the pvMs were considered to be CD45 high.

9. Figure 7: the labeling is not clear: what is pia? It seems too thick for pia mater. How can it be just above the hippocampus? Is there no corpus callosum? The location of the ischemic lesion is confusing. Does the stroke extend into the hippocampus? Which markers beside lyve1 do the cells express after stroke?

The maximum intensity projection was more than 2000µm, meaning that the rounding of the brain becomes flattened and thus that the pia mater appears larger because of the flattening. We mentioned this clearer in the legend. Also, we have added the 3D reconstruction videos of the stroke lesion as Supplementary videos 5 and 6 to better understand curvature and structure of the area involved. Also, to facilitate the understanding of the anatomy, we have added the regions in the section figures. The hippocampus should not be directly affected as can be seen in the attached stroke histology figure to this letter (Fig. 3 attached). The corpus callosum is ending slightly more caudally, so it is partly present but difficult to indicate because it is not stained in the whole-mount staining.

We have analyzed the Lyve1 pvMs also by flow-cytometry, using similar gating as in Fig.3, i.e. Lyve1⁺F4/80⁺ and made a distinction on Cx3cr1 expression.

Fig. 3: Cresyl violet staining of a photothrombotic stroke area in a 50 μm transversal section, indicating a lesion which extends close to the corpus callosum above the hippocampus. The corpus callosum ends slightly more caudally, where also the pia-mater invaginates towards the 3rd ventricle.

Reviewer #3 (Remarks to the Author):

In the current study by Siret et al, the authors describe a perivascular macrophage population positive for Lyve1 but negative for common macrophage markers including non-parenchymal brain macrophages like Cx3cr1, CD206 and CD163. These non-conventional perivascular macrophages (pvM2) were associated to vessels. While they expressed F4/80, they were not positive for PU.1 but needed it for their differentiation. Using Vavtdt mice, pvM2s were not labelled and therefore not of hematopoietic origin.

This is an interesting study and the identification of an additional non-conventional macrophage population associated to the neurovascular unit would be very important for the field.

However, the conclusions are not supported by their data and there are several issues with the current manuscript that need to be addressed.

The evidence is not convincing that these Lyve1⁺ cells are indeed macrophages:
 - The ontogeny part was only addressed with one model (Vav1Cre), suggesting that they are in fact non-hematopoietic cells.

As replied before to reviewer 1, we have addressed their origin with different models. In the current version of the manuscript, using the *Cx3cr1^{Cre}*-reporter we have additionally shown that *Cx3cr1* expression was present within this lineage at an earlier point in their ontogeny (Fig. 5c and Supplementary Fig. 5c). This in line with the data on the *Spi1^{GFP}* line, in which no Lyve1⁺ pvMs were observed (Fig. 5a-b) indicating a dependence on PU.1 during their ontogeny.

Vav1 was described to be expressed only in embryonic hemogenic endothelial descendants, after E11.5, and subsequently in HSC (and all cells derived from the bone-marrow). We have used exactly the same *Vav1^{Cre}* line (B6.Cg-Commd10Tg(Vav1-icre)A2Kio/J) as was previously used to separate between yolk-sac vs other sources of microglia progenitors (Fehrenbach et al., AnnAnat 2018, PMID 29704636).

The *Vav1* tracing model is raising doubts with the reviewer on its specificity and we agree that there are better models to trace the lineage of the pvMs. Therefore, we have now replaced this data with data from the other more specific reporter models. These include the *Cxcr4^{CreErt2};Rosa26^{tdTomato}* reporter mouse model to specify their replacement from the bone-marrow derived monocytes and the *Cx3cr1^{Cre};Rosa26^{tdTomato}* reporter mouse model, next to the already shown data on the *Spi1^{GFP}* reporter mouse model, to determine a common macrophage origin.

It has been recently unequivocally shown that pvMs descent from hemogenic progenitors in the yolk-sac (Masuda et al., Nature 2022, PMID 35444273), similar as the (tissue resident) macrophage lineage which are Vav1 negative. However, in the Masuda et al. publication, there was no clarification on the heterogenous nature of pvMs in adult mouse brains. Using the *Cxcr4^{CreErt2};Rosa26^{tdTomato}* revealed that this subset is indeed not a monocyte derived macrophage in the naive situation (confirming the Masuda et al. publication). Moreover, using the same model, we established that during in photothrombotic stroke the Lyve1⁺ pvMs are not replaced by bone-marrow derived monocytes but are most likely derived from resident cells.

- The RNA-seq. data appeared to be performed on CD45- and Mrc1-expressing cells and does not unequivocally reveal the pvM2 population

We have provided details on the single cell sequencing data and apologize for the lack of it in the initial version. Initially, we showed the *Mrc1* highest expressing cluster of a larger dataset. This doesn't imply all cells within this cluster express *Mrc1*. We mentioned not clear enough that the expression of the *Cx3cr1⁺* or *Cx3cr1⁻* pvM profiles are very similar, and thus did not segregate.

To be concise, we re-analyzed the complete dataset with the different ages of the mouse brains and now show the presence of the Lyve1 vs the MHCII clusters in the complete integrated dataset. Moreover, we have indicated the macrophage expression markers in a dot-plot for the pvM/BAM and microglia. Upon isolating the cluster with the highest *Lyve1* expression (instead of *Mrc1*, excluding the MHCII⁺ macrophages), we cannot segregate the *Lyve1* cluster further based on *Ptpnc* (CD45) or *Cx3cr1*, indicating that the *Cx3cr1⁺* or *Cx3cr1⁻* pvMs are in fact similar cell types. This was also a reason not to mention pvM2 anymore, but annotate these cells in the current manuscript as CX3CR1⁻ pvM.

- While they show nice images, they mostly only show Lyve1+ cells in combination with markers for other cell types. Lyve1 alone is not sufficient to identify macrophages or even distinguish between pvM1 and pvM2s

As the reviewer rightly pointed out, the discriminatory power of immunostaining is more limited compared e.g. to flow cytometry as only a small number of two up to four markers can be analyzed in parallel. Nevertheless, we have attempted to show combinations of the most relevant markers like *Cx3cr1*, PU.1 in combination with F4/80 and Iba1. Our conclusions were certainly not based on only *Lyve1* expression, but on co-localization, and, an updated flow cytometry panel.

- their flow cytometry data is not of sufficient quality

We have redone the flow-cytometry panel and included different macrophage markers. Also, we have compared the Lyve1⁺ pvMs with the microglia.

In more detail:

- Throughout the study, no quantifications are shown for their images. They mostly show 1-2 cells within a region and draw their conclusions based on a couple of cells. No information is provided about how many brains/images were analysed or how many times an experiment was repeated. They would need to show a quantification for each staining and region.

We apologize for not stating the number of experiments done for the immunofluorescence on sections. We have now stated the number of brains analyzed for the representative images shown in the figures. We are convinced that quantification on sections is easily biased mainly due to technical and orientation artifacts. We did quantify accurately the number of the CX3CR1⁺ or CX3CR1⁻ pvMs by flow-cytometry. Second, using whole mount analysis we quantified all Lyve1⁺ pvMs in the complete stroke area, rather than selected areas in selected sections (Fig. 7 and new Supplementary videos 5 & 6). Therefore, we trust our conclusions are solidly based on a significant cell numbers, including the combination of immunofluorescence on several sections from different brains, quantification by flow-cytometry, whole mount imaging and single cell RNA sequencing data.

- They suggest that two populations of Lyve1+ pvMs are present, the conventional (Cx3cr1+) and the non-conventional (Cx3cr1-) ones. What is the percentage of Lyve1+ cells among pvMs that are Cx3cr-?

We thank the reviewer for raising this important point. Even though we focus on a description of the Lyve1⁺ pvM, there is certainly a Lyve1⁻ pvM population present. Indeed, in the flow-cytometry panel, about 35% of the CX3CR1⁻ cells are Lyve1⁺, while 68% of the CX3CR1⁺ cells are Lyve1⁺. This was also clear from the single cell RNA sequencing data, in which a *Lyve1*⁻ subcluster was present (Fig. 4d, supplementary Fig. 4b). This indicates a clear plasticity of the pvMs in expression of the markers. We focused on the Lyve1⁺ cells because this was considered the major part of the pvM population within the brain (Faraco et al., JCI Insights 2016, PMID 27841763) and we wanted to determine if there was a Lyve1⁺ lymphatic endothelial cell presence within the brain

Most stainings only show Lyve1 and they would need additional markers to identify them as conventional pvMs or pvM2 (for example Cx3cr1, CD206, CD163 etc.). Furthermore, they would need to show that the Lyve1+ cells are indeed macrophages by using additional stainings (F4/80 or Iba1) (Fig. 1G, H, I, Fig. 2, Fig. 5, Fig. 6, Suppl. Fig. 1, Suppl. Fig. 2).

For this important point, we would like to refer to Fig. 1e-g, in which we establish that all parenchymal and vessel-associated Lyve1⁺ cells in sections also express F4/80, Iba1 and CD206. Although it was technically not possible to include markers in the other figures, we are convinced that the results in Fig. 1 are highly representative for the identity of the pvM. This is also backed-up by the added Supplementary figure 5b in which we now show that in a PU.1^{GFP/+} brain all Lyve1⁺ cells are also F4/80⁺ and CD206⁺, regardless of PU.1 expression. In Fig. 2, we used lymphatic markers to establish whether Lyve1+ cells could be lymphatic endothelial cells, which we strongly excluded. Furthermore, Fig. 2. is concerning location of the Lyve1+ cells within the perivascular space. We have added immunofluorescence stainings showing the localization of the pvMs within the peri-vascular space in great detail.

- Overall, it would be nice to include DAPI, at least in the higher magnification images.

We thank the reviewer for this suggestion and have added DAPI staining to Fig. 2g and also added supplementary videos to establish that the stretched cells contain nuclei (Fig. 2g, Supplementary Videos 3&4).

- In Figure 1, the authors describe that Lyve+Cx3cr1(GFP)- Iba1+F4/80+ macrophages lack CD45 and CSF-1R. However, the expression of both these markers is difficult to assess by immunohistochemistry. For example microglia and also some non-parenchymal

macrophages are typically CD45^{lo}, which cannot be detected by histology, as evidenced by Fig. 1G where microglia (CD45^{lo}) are not positive for CD45. Thus, the statement that pvM2 are CD45 negative cannot be made by this image.

Indeed, we thank the reviewer for bringing up this important matter. Marker expression can be difficult to show on sections, especially CD45 is notoriously difficult to stain on microglia in brain sections (e.g. Fig. 1h, Goldmann et al., Nat.Imm. 2016, PMID 27135602). In our observation the conventional Lyve1⁺CX3CR1⁺ pvMs are CD45 positive, also in section stainings, while the Lyve1⁺CX3CR1⁻ pvMs have much lower expression or undetectable CD45 expression in immunofluorescence staining on sections. CD45 expression on microglia is normally detected by cytometry. As shown in the new flow-cytometry data, we compared the Lyve1⁺ pvMs to the microglia (Supplementary Fig. 3b). While CD45 expression in immunofluorescence on sections was not detectable on Lyve1⁺CX3CR1⁻ pvM, similar as for microglia, we observed a CD45 low to intermediate population in flow-cytometry, expressing CD45 at similar levels as microglia (Supplementary Fig 3b). Therefore, we agree with reviewer that we cannot exclude that these cells are not negative for CD45, so we now refer to the Lyve1⁺CX3CR1⁻ pvMs being CD45 low to intermediate. Accordingly, we have adjusted the title of the manuscript. We like to note that earlier publications on CD45 expression on brain pvMs state that pvMs express CD45 at high levels, while we observed similar levels as for microglia. This information is important in order to setup pvMs gating strategies in future studies.

Regarding CSF-1R, the authors show one Lyve1⁺ cell, which is negative for CSF-1R. However, this one specific cell highlighted with a red arrow appears to be very weakly positive for Lyve1 and whether this cell is indeed a cell and a pvM is not clear. As mentioned above, an additional staining such as F4/80 or Iba1 would be helpful and also Dapi. Thus, according to their image, it cannot be claimed that Lyve⁺Cx3cr1⁻ pvMs lack CSF-1R expression.

We have addressed the Dapi and Csf1R immunofluorescence staining in collaboration with the lab of Lydia Sorokin. We could not improve the quality of the Csf1R immunofluorescence staining, but detected signal in the flow cytometry (Fig. 3a). Hence, we have removed the Csf1R immunofluorescence data and statement. The Dapi staining, as requested in a previous question, is shown in Fig. 2g and Supplementary videos 3&4.

For their flow cytometry data:

- They claim that microglia were excluded as they are Lyve1⁻. However, it seems that microglia are still included (Fig. 3) and that the two populations expressing F4/80, GFP and intermediate levels of CD45 are microglia and non-parenchymal macrophages. Thus, Lyve1 does not look like a convincing staining and it is therefore absolutely critical that they show an isotype for Lyve1 for the different cell types and ages.

In the new flow-cytometry panel (Fig. 3a and Supplementary Fig. 3), we now compare the Lyve1⁺ pvMs with the microglia. Also, from Supplementary Fig. 3b, it is clear that the microglia are indeed Lyve1 negative. We have used an FMO (grey in the histograms) plus added the isotype control for Lyve1 (see attached figure 3). The use of other macrophage markers like CD64 and CD206 in cytometry rule out microglia contamination.

Fig. 3: Isotype control for anti-Lyve1 on dissociated single, live cells isolated from adult mouse brains.

Other markers should be used in addition to identify microglia and non-parenchymal macrophages (for example CD64), and monocytes/neutrophils should be excluded. They should gate first on total macrophages and then show Lyve1 expression and the other markers.

We have used a new cytometry panel, now including CD64, CD206, Csf1R and CD163. Next to the immunofluorescence and single cell RNA sequence data, these panels clearly show a macrophage identity of the Lyve1⁺ cells (Fig. 3), while the microglia are Lyve1 negative (Supplementary Fig. 3b). We also added histology analysis on the sorted Lyve1⁺ macrophages (Fig 3c) to indicate these cells are bona-fide macrophages.

- In addition, 60-80% of the CD45-Lyve⁺ cells are F4/80 negative (Figure 3). This clearly indicates that most CD45-Lyve1⁺ cells are NOT macrophages and emphasizes again that Lyve1 alone does not suffice to classify these cells as macrophages in the immunohistochemistry analysis.

In Fig.1 e-g, Supplementary Fig. 5b we established by immunofluorescence on sections that all Lyve1⁺ cells were F4/80, CD206 or Iba1⁺. These results show that the parenchymal vessel associated Lyve1⁺ cells are almost exclusively F4/80, Iba1 or CD206⁺. Based on the new panel and the comparison to microglia therein, we agree to tune down the statement on CD45 negative Lyve1⁺ cell, but like to state that the expression of CD45 is comparable to microglia, being low to intermediate, which is certainly lower as stated in previous pvM publications.

- Figure 3E-F, why would they gate through the middle of the F4/80 population? These are clearly all F4/80⁺ cells.

We have redone the cytometry panels and have carefully drawn the gates. All gates have been setup using the FMO or isotype control (Lyve1)(see page above). The F4/80 expression on these cells does change during aging (Fig. 3a) and after P21, the F4/80 is clearly separated.

scRNA-seq data:

- It is not clear which cells were sequenced. In the material methods section, they state that for the RNA-seq data, they extracted Mrc1-expressing cells for analysis. However, in Supplementary Figure 1D, they claim that the pvM2 do not express CD206, meaning that their pvM2 population would not be included. On the other hand, they also write that the sequencing was performed on non-neuronal CD45⁺ cells. However, most parenchymal

macrophages are CD45^{lo} and would therefore have been excluded. (as stated above, they do not convincingly demonstrate that pvM2 are indeed CD45⁻).

We have updated all the information regarding the single cell RNA sequencing and now show the presence of the *Lyve1*⁺ cluster within the complete dataset, rather than selecting for *Mrc1* (CD206). Also, the dataset is used in another manuscript under consideration with eLife (BioRxiv <https://doi.org/10.1101/2022.06.10.495613>). The focus of this study was the role of the endothelial cells in aging and during EAE and on MHCII⁺ pvM/BAM during EAE. Our study concerns the description of the *Lyve1*⁺ pvM cluster. However, if required, the single cell RNA sequencing data can be checked in this pre-print in more detail, especially for the clusters we ignored. Unfortunately, the comment “they also write that the sequencing was performed on non-neuronal CD45⁻ cells. However, most parenchymal macrophages are CD45^{lo} and would therefore have been excluded. “ is not based on information we supplied before in the manuscript. We would like to stress that we did not select for CD45⁻ cells, or that we excluded CD45⁺ cells. In the previous version of the manuscript, we mentioned “we analyzed the single cell RNA sequencing dataset of the non-neuronal cell population in mouse brain cortex (GSE133283), thus including the CD45⁻ cells (Fig. 4B, C).”. The source of the single cell RNA sequencing dataset has been explained clearer in the current version.

In Figure 4B, the two clusters are not annotated. What are they supposed to be? Conventional pvM and pvM2 (red)? Yet, the cells expressing *Lyve1*, also express *Spil1* (whereas in their reporter strain, pvM2 did not express *Spil1*). Also, *Cx3cr1* seems to be expressed ubiquitously. Overall, the RNA-sequencing data did not reveal a cluster/population that corresponds to pvM2s.

We have completely redone the single cell RNA sequencing analysis. In this analysis, we clearly show only 1 *Lyve1* expressing cluster within the complete dataset, which expressed macrophage markers as shown in the dotplot in Fig. 4c. Subsequently, we isolated this cluster for a detailed analysis on a possible segregation based on *Cx3cr1* or *Ptprc*. Indeed, the *Lyve1*⁺ pvM cluster cannot be segregated based on *Cx3cr1* or *Ptprc* expression profile. In the previous version of the single cell RNA sequencing data, we showed a cluster which included both *Lyve1* and MHCII expressing cells. We mentioned that the segregation between MHCII vs *Lyve1* perivascular macrophages was observed in the periphery before (Chararov et al., Science 2019, PMID 30872492). Our data (both single cell RNA sequencing as well as immunofluorescence (Fig.1h)) confirm this segregation. The segregation is also visualized in the new version as the *Lyve1*⁺ cluster is segregated from the *H2-Aa* (MHCII) cluster (Fig. 4b, Supplementary Fig. 4a).

- *Cx3cr1* was not expressed ubiquitously. The low vs higher expressing cells (and to show non-ubiquitous expression) is illustrated in the violin plots in Supplementary Fig. 4a.

- *Spi1*^{GFP/+} reporter model indeed showed the lack of PU.1 (GFP) expression within some of the *Lyve1*⁺*CX3CR1*⁻ pvM. However, RNA transcripts were indeed present within the *Lyve1*⁺ cluster. They could still have the *Spi1* mRNA transcripts, but not the protein or GFP reporter as the quantity of transcripts vs. protein is not necessarily the same.

Summarized, the main point of our study is that the pvM population is heterogeneous population, which has been previously disregarded. Especially the lack of *CX3CR1* in about 15% of the pvMs and PU.1 in some, plus the lower CD45 expression is important to realize to analyze the pvM population.

- A transcriptome analysis of pvM2 would be critical to show that they are indeed macrophages and what markers they express.

We have re-analyzed the complete dataset and have now provided a dot-plot with macrophage or microglia associated genes of the *Lyve1*⁺ specific cluster compared to the MHCII⁺ pvM/BAM and microglia (Fig. 4c). Within the *Lyve1* highest expressing cluster, there is no segregation based on *Cx3cr1* or *Ptprc* but rather on *Lyve1*, *Cd209* and the Fos complex associated genes. It indicates another level of complexity of this population, but not based on *Cx3cr1* or *Ptprc* expression.

- **Vav1 fate-mapping: they demonstrate that the *Lyve1*⁺ cells are tdtomato-. Again, quantification needs to be added. How many of the non-conventional pvMs are dtTomato-, in comparison to microglia and the *Lyve1*⁺ conventional PVMs for example? The fact that the 'pvM2' are of non-hematopoietic origin suggests that these are not macrophages.**

As we have described before to this reviewer on page 9-10, *Vav1* is expressed in the embryonic hemogenic endothelium and not in yolk-sac derived hemogenic endothelial cells. We are sorry that we were not clearer before that lack of *Vav1* expression doesn't mean non-hematopoietic. To prevent any misunderstanding, we have replaced this model by more specific reporter models which support the data of another publication (Masuda et al., Nature 2022).

An additional model would be important to demonstrate this. What is their ontogeny?

As indicated in our reply to reviewer 1 points 1-3, we have now used the *Cx3cr1*^{Cre};*Rosa26*^{tdTomato} illustrate a macrophage origin, plus the *Cxcr4*^{CreErt2};*Rosa26*^{tdTomato} model to establish the *Lyve1*⁺ pvMs are not bone-marrow monocyte derived macrophages in naïve conditions and in stroke.

- **Can pvM2 be found around veins and arteries? Have they looked at capillaries where no perivascular space is observed?**

We have observed in whole mount immunofluorescence staining that the pvMs were not present around non-arterial capillaries (fig 2a). Higher resolution images of their location within the peri-vascular space have been added (Fig. 2g) in collaboration with the lab of Lydia Sorokin, a renowned expert on the peri-vascular space.

- **There is no information about how many images and how many brains were analyzed or how many times an experiment was repeated.**

As mentioned in response to an earlier comment by this reviewer, we have addressed this. For all other experiments, we already stated the number of animals/experiments.

REVIEWER COMMENTS

Reviewer #1 (Remarks to the Author):

I have no further concerns

Reviewer #2 (Remarks to the Author):

The revised version of the paper addresses most concerns raised in the previous review. The following points deserve consideration:

1. Lyve+ cells are also observed in the pia, and their location is not exclusively in the perivascular space. This distinction should perhaps be pointed out and a different abbreviation used.
2. Brain macrophages have also been implicated in the deleterious cerebrovascular effects of amyloid-beta (PMID: 28515043) and in the BBB dysfunction in hypertension (PMID: 32654560), which may deserve mention. Since in these studies clodronate or bone marrow chimeras were used to eliminate or genetically modify brain macrophages, respectively, one wonders which population of perivascular and meningeal macrophages were targeted.
3. There is no justification for why the brain was examined 14 days after photothrombotic stroke. Since there are dynamic changes in infiltrating immune cells, invading monocytes could have been missed. Furthermore, the infiltration is proportional to the size of the stroke lesion (PMID: 33496918), which is small in photothrombotic stroke. These potential confounders may need to be considered.

Reviewer #3 (Remarks to the Author):

They have improved the manuscript but I still have some comments:

They included two additional models for the ontogeny part. The Cx3cr1GFP/Cre tdTom model nicely demonstrates that while all Lyve1+ macrophages are tdTom+, a small population is no longer expressing Cx3cr1 (GFP) but is derived from a Cx3cr1+ precursor. However, as this is not an inducible model, the origin of the Cx3cr1- pvMs has not been defined.

The other model is unclear. In Cxcr4CreER reporter mice all HSC are labeled while embryonically-derived macrophages remain unlabeled upon tamoxifen treatment postnatally. In their experiments, pvMs were analyzed 2 weeks after tamoxifen administration. This time point is not long enough as it would take weeks/months for a macrophage population to be replaced by labeled monocytes; even for populations that were previously described to be replaced over time such as choroid plexus or dural macrophages (see van hove et al. 2019, Goldmann et al. 2016).

Furthermore, it is known that pvMs in the CNS are not replaced by BM-derived monocytes but are embryonically-derived and expand postnatally. Therefore, in Fig. 5e, it is important to show whether the Cx3cr1- pvMs are also not labeled. Thus, Cx3cr1 expression needs to be shown to distinguish both populations; for example by adding a Cx3cr1 antibody to their flow cytometry panel.

The tdTomato labeling would need to be shown for both Cx3cr1+ and Cx3cr1- populations

and as positive and negative controls monocytes and microglia, respectively.
Second panel in Fig. 5e, presumably the y-axis label should be tdTomato and not Cxcr4?

Overall, in Fig. 5e and also in the other flow cytometry experiments, the authors should pre-gate on macrophages (CD45^{int/lo}, CD64 etc., and most importantly gate out monocytes, neutrophils by using Ly6C/Ly6G etc). It would make it much easier where to draw the Lyve1 gate (or alternatively also use CD163 or CD206 instead of Lyve1).

In Fig. 3a and Suppl. Fig. 3b, they mention that the Cx3cr1⁻ population expressed CD45 at low levels. However, at P7, P21 and adult, there are clearly two populations of CD45 (high and low). It is not clear whether monocytes, DCs, neutrophils were gated out. (See above)

It is not clear why in Fig. 3 the Lyve1⁺ (CX3cr1⁺) population is negative for CD163.

CSF1R staining for flow cytometry after enzymatically digested tissue is difficult and usually does not work (as can be seen in their panels).

Other (recent) reports describing the Lyve1⁺ pvMs in the brain should also be cited, such as Zeisel et al. Science 2015, Karam et al, JCBFM 2022.

Please find below a point-by-point rebuttal of the concerns raised by the reviewers. All changes to the text are highlighted in yellow in the manuscript.

Reviewer #1 (Remarks to the Author):

I have no further concerns

We thank the reviewer for the previous suggestions and are happy that all concerns were addressed.

Reviewer #2 (Remarks to the Author):

We thank the reviewer for valuable remarks and suggestions. Please find our answers to the points raised below.

The revised version of the paper addresses most concerns raised in the previous review. The following points deserve consideration:

1. Lyve+ cells are also observed in the pia, and their location is not exclusively in the perivascular space. This distinction should perhaps be pointed out and a different abbreviation used.

Indeed, there are several populations of Lyve1+ cells within the pia mater. As others have shown before and we have shown in this manuscript, these include single lymphatic endothelial cells (Shibata-Germanos et al., Acta Neuropathol. 2019, PMID 31696318; van Lessen et al., eLife 2017 PMID 28498105) as well as Lyve1+ macrophages. To clearly emphasize our study is focused on the parenchymal pvMs, we have added 'parenchymal' on line 81 and 'within the parenchyma' (Line 83). Furthermore, in all cases where this focus of the study might not be fully obvious, we have added this term in lines 83, 86, 89, 100, 107, 113, 122, 134, 138, 147, 153, 203.

2. Brain macrophages have also been implicated in the deleterious cerebrovascular effects of amyloid-beta (PMID: 28515043) and in the BBB dysfunction in hypertension (PMID: 32654560), which may deserve mention. Since in these studies clodronate or bone marrow chimeras were used to eliminate or genetically modify brain macrophages, respectively, one wonders which population of perivascular and meningeal macrophages were targeted.

We apologize for these omissions. We have added the references in the text where we provide background on the function of these macrophages (lines 65- 66).

We have refrained from discussing the differences between meningeal and parenchymal macrophages as we focused on the parenchymal macrophages and did not study the meningeal macrophages in detail. A general concern in the analysis of macrophage ontogeny is that almost all macrophages at some stage in their differentiation path express similar genes, and thus genetic models are notoriously unspecific. Clodronate is even more unspecific, as it targets all phagocytosing cells, including DCs etc. Therefore, the use of calcein loaded liposomes coated with antibodies to target a specific macrophage population could be very promising to study sub-populations (e.g. Etzerodt, JEM 2019, PMID: 31375534). However, as mentioned in the discussion, since we are not aware of a unique marker for the CX3CR1-negative parenchymal pvMs, we could not target these cells specifically in a genetic- or liposome treatment model.

3. There is no justification for why the brain was examined 14 days after photothrombotic stroke. Since there are dynamic changes in infiltrating immune cells, invading monocytes could have been missed. Furthermore, the infiltration is proportional to the size of the stroke lesion (PMID: 33496918), which is small in photothrombotic stroke. These potential confounders may need to be considered.

We have added time points day 3 and 8 after lesion induction (Fig 7d). We noted the highest increase of Lyve1+F4/80+ cells at day 3 and have accordingly deleted the comment that the peak was at day 14

(previously in the discussion on line 285). Monocytes have been shown to be among the first immune cells to enter the lesion and they arrive in great numbers within the first 3 days. However, the analysis of these (Cxcr4⁺) monocytes was described in detail in another publication (Werner et al., NatNeuro 2020, PMID: 32042176), and therefore we have not focused on these cells in our study.

To prevent a bias in the number of the infiltrating cells due to lesion size, we have carefully induced the lesion always at exactly the same position using a stereotactic device and a mold to precisely limit the light beam. Furthermore, induction of the lesion and subsequent isolation of the brain were performed following a closely supervised temporal regime. Therefore, size and location of all photothrombotic lesions should have been similar and the number of infiltrating immune cells should not have been dictated by different tumor sizes.

Reviewer #3 (Remarks to the Author):

We thank the reviewer for valuable remarks and suggestions.

They have improved the manuscript but I still have some comments:

They included two additional models for the ontogeny part. The Cx3cr1GFP/Cre tdTom model nicely demonstrates that while all Lyve1⁺ macrophages are tdTom⁺, a small population is no longer expressing Cx3cr1 (GFP) but is derived from a Cx3cr1⁺ precursor. However, as this is not an inducible model, the origin of the Cx3cr1⁻ pvMs has not been defined. The other model is unclear. In Cxcr4CreER reporter mice all HSC are labeled while embryonically-derived macrophages remain unlabeled upon tamoxifen treatment postnatally. In their experiments, pvMs were analyzed 2 weeks after tamoxifen administration. This time point is not long enough as it would take weeks/months for a macrophage population to be replaced by labeled monocytes; even for populations that were previously described to be replaced over time such as choroid plexus or dural macrophages (see van hove et al. 2019, Goldmann et al. 2016).

Furthermore, it is known that pvMs in the CNS are not replaced by BM-derived monocytes but are embryonically-derived and expand postnatally. Therefore, in Fig. 5e, it is important to show whether the Cx3cr1⁻ pvMs are also not labeled. Thus, Cx3cr1 expression needs to be shown to distinguish both populations; for example by adding a Cx3cr1 antibody to their flow cytometry panel. The tdTomato labeling would need to be shown for both Cx3cr1⁺ and Cx3cr1⁻ populations and as positive and negative controls monocytes and microglia, respectively. Second panel in Fig. 5e, presumably the y-axis label should be tdTomato and not Cxcr4?

We have addressed these concerns and repeated the analysis on the inducible Cxcr4CreErt2;Rosa^{tdTomato} reporter model. We have now injected tamoxifen 4 weeks before the analysis. Furthermore, we have added the CX3CR1 antibody to the cytometry staining panel for analysis of the Cxcr4CreErt2;Rosa^{tdTomato} induced mice to trace the lineage of both CX3CR1⁺ and CX3CR1⁻ pvMs (Fig. 5e, lines 198-202). Both pvM populations showed a similar dependence on local progenitors and were not replaced by bone-marrow-derived monocytes. The lineage tracing data were verified by the analysis of microglia and monocytes and/or neutrophils (Supplementary Fig. 5e, lines 200-202). Microglia were not labeled by tdTomato, while the monocytes/neutrophils were to a larger extent, thus validating the Cxcr4CreErt2;Rosa^{tdTomato} model used.

Overall, in Fig. 5e and also in the other flow cytometry experiments, the authors should pre-gate on macrophages (CD45int/lo, CD64 etc., and most importantly gate out monocytes, neutrophils by using Ly6C/Ly6G etc). It would make it much easier where to draw the Lyve1 gate (or alternatively also use CD163 or CD206 instead of Lyve1).

Since one aim of this manuscript is to show an intermediate CD45 expression of the pvM opposed to CD45^{high} expression stated in other publications, we did not pre-gate on this marker and show its expression in this manuscript as a histogram. Indeed, as stained sections and single cell RNA sequencing data confirmed that all Lyve1⁺ pvMs expressed F4/80, CD64 and CD206, we have used these markers to pre-gate the pvMs (Fig. 3 and Supplementary Fig. 3). Using the suggested Ly6C and

Ly6G markers, we have ruled out that the F4/80⁺CD64⁺CD206⁺ pvMs are monocytes or neutrophils as they were double negative for both (Supplementary Fig. 3a).

In Fig. 3a and Suppl. Fig. 3b, they mention that the Cx3cr1⁻ population expressed CD45 at low levels. However, at P7, P21 and adult, there are clearly two populations of CD45 (high and low). It is not clear whether monocytes, DCs, neutrophils were gated out. (See above)

It is not clear why in Fig. 3 the Lyve1⁺ (CX3cr1⁺) population is negative for CD163.

CSF1R staining for flow cytometry after enzymatically digested tissue is difficult and usually does not work (as can be seen in their panels).

To our knowledge, there are no publications on DCs or neutrophils expressing Lyve1 and thus we are not aware of any other immune cell expressing Lyve1 besides specific macrophages. Therefore, in the initial cytometry panels, gating of F4/80⁺Lyve1⁺ double positive cells effectively rules out a contamination by DCs, monocytes or neutrophils. However, to specifically address the reviewer's concerns and to validate our approach using the suggested strategy, we also applied gating for F4/80⁺CD64⁺CD206⁺ positive cells as requested by the reviewer and confirmed that these are negative for both Ly6C and G. Therefore, the probability that the cells shown in these plots are not macrophages is extremely small. Indeed, at neonatal stages there are very few CX3CR1⁻ pvMs, and this inherently results in multiple peaks. Also, the presence of differentiating progenitors within very young animals could result in increased heterogeneity of the population.

We agree on the Csf1R staining problem and thank the reviewer for raising this issue. Moreover, we are aware that this receptor is internalized especially during neonatal stages when the progenitors are sensitive to Csf1 signaling. Therefore, we concur with the reviewers argument and have removed this staining as it might not reflect the presence of Csf1R in or on the cells.

The CD163 expression in the CX3CR1⁺ population is indeed low. However, it is above the background control/FMO. Interestingly, CD163 indeed higher in the CX3CR1⁻ population, raising the question why these cells have downregulated CX3CR1 and seemingly upregulated CD163. To our knowledge, CD163 expression on brain parenchymal pvMs has not been studied in neonates before. Another very recent study showed that before neonatal day 10 (P10) exclusively pvM progenitors were present and only after P10 the first mature pvMs were observed (Masuda et al., Nature 2022, PMID: 35444273). Therefore, CD163 might not be present on progenitors but expression may initiate in mature pvMs, and become upregulated on CX3CR1⁻ pvMs.

Other (recent) reports describing the Lyve1⁺ pvMs in the brain should also be cited, such as Zeisel et al. Science 2015, Karam et al, JCBFM 2022.

We regret that these references were not included and have added them (line 76). We have discussed the study of Karam et al on the parenchymal Lyve1⁺ pvM staining (line 78-79) and their presence on specific veins and all arteries (line 259-261).

REVIEWERS' COMMENTS

Reviewer #2 (Remarks to the Author):

The authors have addressed my concerns successfully.

Reviewer #3 (Remarks to the Author):

The authors have now addressed my comments.